# SAFER: A Calibrated Risk-Aware Multimodal Recommendation Model for Dynamic Treatment Regimes

Yishan Shen [*1]   Yuyang Ye [*2]   Hui Xiong [3]   Yong Chen [1]

## Abstract

Dynamic treatment regimes (DTRs) are critical to precision medicine, optimizing long-term outcomes through personalized, real-time decision-making in evolving clinical contexts, but require careful supervision for unsafe treatment risks. Existing efforts rely primarily on clinician-prescribed gold standards despite the absence of a known optimal strategy, and predominantly using structured EHR data without extracting valuable insights from clinical notes, limiting their reliability for treatment recommendations. In this work, we introduce SAFER, a calibrated risk-aware tabular-language recommendation framework for DTR that integrates both structured EHR and clinical notes, enabling them to learn from each other, and addresses inherent label uncertainty by assuming ambiguous optimal treatment solution for deceased patients. Moreover, SAFER employs conformal prediction to provide statistical guarantees, ensuring safe treatment recommendations while filtering out uncertain predictions. Experiments on two publicly available sepsis datasets demonstrate that SAFER outperforms state-of-the-art baselines across multiple recommendation metrics and counterfactual mortality rate, while offering robust formal assurances. These findings underscore SAFER's potential as a trustworthy and theoretically grounded solution for high-stakes DTR applications.

*Equal contribution [1]University of Pennsylvania [2]Rutgers University [3]The Hong Kong University of Science and Technology (Guangzhou). Correspondence to: Hui Xiong <xionghui@ust.hk>, Yong Chen <ychen123@pennmedicine.upenn.edu>.

*Proceedings of the 42$^{nd}$ International Conference on Machine Learning*, Vancouver, Canada. PMLR 267, 2025. Copyright 2025 by the author(s).

## 1. Introduction

How can we enable models to recognize when they are uncertain about their predictions? Safely providing personalized, sequential treatment recommendations that adapt to a patient's evolving clinical state is a longstanding challenge in optimizing outcomes in high-stakes healthcare scenarios. In this work, we address this challenge within the framework of dynamic treatment regimes (DTRs) (Robins, 1986; Murphy, 2003; Chakraborty & Moodie, 2013; Tsiatis et al., 2019). Crucial for real-world decision-making, resource allocation, and reducing trial-and-error treatments (Murphy, 2005; Laber et al., 2014), DTR requires more precise, adaptive, and safe control over treatment strategies while minimizing risks in critical clinical contexts.

Recent advances in deep learning (DL) have significantly improved DTR frameworks by addressing key challenges such as patient heterogeneity, temporal dependencies, and the high-dimensional clinical data (Kosorok & Laber, 2019; Moodie et al., 2007). DL offers distinct advantages for DTR, including the ability to integrate heterogeneous data sources, such as electronic health records (EHRs) and temporal patterns, while capturing complex dependencies over time (Yu et al., 2021; Ching et al., 2018; Olawade et al., 2024; Melnychuk et al., 2022). However, a major challenge in current DTR approaches is the absence of optimal treatment strategies, particularly for understudied diseases, critically ill or deceased patients as their outcomes may not reliably indicate the most appropriate clinical actions (Robins et al., 2000; Schulam & Saria, 2017; Chapfuwa et al., 2021). This inherent label uncertainty in DTR remains underexplored, limiting the robustness of existing models. Furthermore, most approaches lack theoretical guarantees on their recommendation quality, leaving practitioners without principled mechanisms for error rate control (e.g. Benjamini & Hochberg, 1995; Lei et al., 2018; Bates et al., 2023; Jin & Candès, 2023).

Additionally, many existing methods (e.g., Murphy, 2005; Laber et al., 2014; Bica et al., 2020) rely primarily on structured EHR data while underutilizing the valuable textual information contained in clinical notes, which often capture critical insights into a patient's history and physician assessments. However, integrating clinical notes into DTR

remains a challenge due to difficulties in establishing a unified embedding space that preserves inter-modality context, temporal alignment, and domain semantics. Prior work, such as Choi et al. (2017); Shang et al. (2019a), has explored representation learning for medication recommendation, but these approaches still rely heavily on medical code representations derived from structured EHR data.

We address these challenges through two key desiderata: (i) uncertainty control—the DTR model should quantify prediction uncertainty and provide statistically guaranteed control over the uncertainty discovery threshold specified by the user, and (ii) comprehensive information fusion—the model should integrate all available patient data, ensuring no critical information is overlooked. Together, we refer to these principles as "risk awareness".

In this work, we propose SAFER, a Calibrated Risk-Aware multimodal framework for DTR that enhances the robustness of DTR frameworks. The overall pipeline is depicted in Figure 1. SAFER introduces several key innovations:

**1. Multimodal Representation Learning.** SAFER integrates both structured EHR data and unstructured clinical notes using a novel Transformer-based architecture to learn a unified sequential patient representation. A self-attention mechanism captures inter-modality temporal dependencies, while cross-attention extracts contextual information across modalities (Section 4.1).

**2. Uncertainty-Aware Training.** SAFER accounts for label uncertainty by assuming ambiguous treatment labels particularly for deceased patients. By recognizing that label uncertainty is systematic and predictable, we introduce an uncertainty quantification module that assigns per-label risk scores and incorporates them into a risk-aware loss function for interactive training (Section 4.2).

**3. Theoretical Guarantees on Prediction Reliability.** Given the critical need for error control in high-stakes scenarios, We derive theoretical guarantees on calibrated recommendations by innovatively employing a conformal inference framework (Vovk et al., 2005; Benjamini & Hochberg, 1995) to control the expected proportion of unreliable predictions (i.e., FDR) at decision time (Section 5).

**4. Empirical Validation.** We evaluate SAFER on real-world EHR benchmarks, demonstrating consistent improvements over state-of-the-art DTR methods across multiple recommendation metrics and reductions in counterfactual mortality rates.[1]

Together, these advancements establish SAFER as a multimodal, risk-aware DTR framework with strong theoretical foundations and superior empirical performance.

---

[1]Our code and dataset are avaliable at https://github.com/yishansssss/SAFER.

## 2. Related Works

**Dynamic treatment regimes.** Prescribing medications in response to the dynamic states of patients is a challenging task. Over the past decade, to model complex, high-dimensional, and temporal healthcare data, researchers have leveraged various deep learning-based approaches to improve treatment recommendations, including RNNs and their variants (e.g., Choi et al., 2016; Bajor & Lasko, 2017; Jin et al., 2018), attention networks and transformer-based models (e.g., Peng et al., 2021; Wu et al., 2022), deep reinforcement learning (DRL) techniques (e.g., Bothe et al., 2013; Komorowski et al., 2018; Raghu et al., 2017; Saria, 2018; Wang et al., 2018; Zhang et al., 2017), convolutional neural networks (CNNs) (e.g., Suo et al., 2017; Cheng et al., 2020; Su et al., 2022), and generative adversarial networks (GANs) (e.g., Wang et al., 2021a;b). Despite these advancements, assessing the effectiveness and ensuring reliable inference for data-driven DTR approaches remains a significant challenge due to variability in evaluating the quality of suggested prescriptions (Hussein et al., 2012; Chakraborty et al., 2014). For instance, while DRL excels in learning optimal DTRs by discovering dynamic policies, inconsistencies in reward design, policy evaluation, and MDP formulations often hinder standardized and rigorous healthcare applications (Luo et al., 2024). We hypothesize that these challenges can be mitigated by modeling predictive uncertainty and generating selective candidates via conformal inference, while providing statistical guarantees.

**Risk-aware treatment recommendation.** To the best of our knowledge, this work is the first to incorporate uncertainty modeling and employ conformal prediction (CP) into DTR research. Several studies have integrated drug-drug interaction knowledge to optimize personalized medication combinations and minimize adverse outcomes. (Shang et al., 2019b; Wu et al., 2022; Tan et al., 2022; Yang et al., 2021). In contrast, we incorporate an uncertainty quantification module to improve both the accuracy and safety of DTRs, ensuring statistically reliable treatment recommendations. As a pivotal role in optimization and decision-making process, prior work has utilized uncertainty quantification in computer vision (Liu et al.; Harakeh et al., 2020), image/video restoration (Shao et al., 2023; Dorta et al., 2018), natural language processing (Chen et al., 2015; Lin et al., 2023; Ren et al., 2023), bioinformatics (Xia et al., 2020; Bian et al., 2020), etc. In the healthcare domain, (Chua et al., 2023; Liu et al., 2024) exemplifies recent efforts to address prediction uncertainty in clinical machine learning models. While our framework is specifically designed for DTR prediction with rigorous theoretical guarantees via CP, we emphasize a complementary but distinct focus on label uncertainty in dynamic treatment regimes. We draw further inspiration from selective prediction, particularly CP-based selection (Bates et al., 2023; Jin & Candès, 2023; Gui et al.,

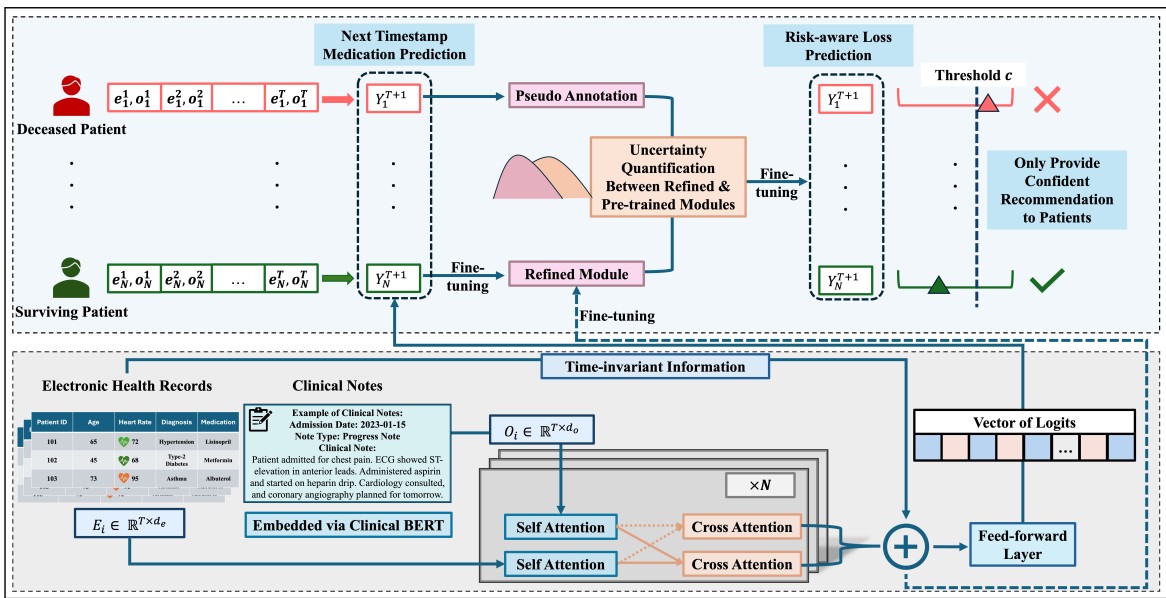

Figure 1. The overall framework of SAFER.

2024), to develop an end-to-end safe DTR framework with formal assurances.

**Clinical notes combined with EHR.** Clinical notes, central to patient care, capture physicians' thought processes, observations, and treatment rationale often absent from structured EHR data (Sheikhalishahi et al., 2019; Rosenbloom et al., 2011). Integrating clinical notes with structured EHRs has demonstrated improved predictive performance in biomedical research, enabling more comprehensive patient modeling (Gao et al., 2024; Lyu et al., 2023). However, current DTR research largely ignores clinical notes due to challenges like data heterogeneity, unstructured text processing, and the absence of standard tools, with most representation learning frameworks relying on structured medical codes (Choi et al., 2017; Shang et al., 2019b). This work bridges these gaps by incorporating clinical notes into DTR frameworks through a novel transformer-based multimodal fusion approach to enhance decision-making accuracy and reliability.

## 3. Problem Setup

### 3.1. Risk-aware Dynamic Treatment Prediction

We aim to model dynamic treatment prediction using two complementary data sources: structured electronic health records (EHR) $\mathcal{E} = \{\mathbf{E}_1, \mathbf{E}_2 \ldots, \mathbf{E}_N\}$ and unstructured clinical notes $\mathcal{O} = \{\mathbf{O}_1, \mathbf{O}_2 \ldots, \mathbf{O}_N\}$. For each patient $i$, where $i \in \{1, \ldots, N\}$, structured data $\mathbf{E}_i$ are represented as a sequence of tabular records $\{\mathbf{e}_i^1, \mathbf{e}_i^2, \ldots, \mathbf{e}_i^T\}$, while unstructured data $\mathbf{C}_i$ consist of a sequence of clinical notes $\{\mathbf{o}_i^1, \mathbf{o}_i^2, \ldots, \mathbf{o}_i^T\}$. Each vector $\mathbf{e}_i^t \in \mathbb{R}^{d_{\mathcal{E}}}$ represents the tabular features at time step $t$, where $t \in 1, \ldots, T$ and $d_{\mathcal{E}}$

denotes the dimensionality of the tabular covariates. Similarly, $\mathbf{o}_i^t$ corresponds to the clinical note associated with patient $i$ at time step $t$. By integrating these two modalities, we aim to predict the medications to be administered in the next clinical decision window, i.e., time step $T + 1$, enhancing treatment recommendations with both numerical clinical metrics and rich textual context.

A significant challenge in DTR lies in uncertainty associated with treatment labels, particularly for negative trajectories (i.e., patient states resulting in adverse outcomes). Specifically, patients with negative outcomes during hospital stay often exhibit unstable and irregular medication patterns, leading to label ambiguity. While some studies in recommendation disregard negative trajectories (e.g., Sun et al., 2021; Ye et al., 2024; 2025), others, like Wang et al. (2020), incorporate this information to refine learned policies and avoid repeating errors. Similarly, we retain negative trajectories but attribute their ambiguity to two primary cases: (1) appropriate treatments were administered but were insufficient to prevent death, and (2) the likelihood that incorrect treatments contributed to adverse outcomes. In contrast, patients with positive outcomes generally display labels that more reliably represent effective clinical decisions. We adopt above assumptions throughout this work.

We propose addressing label uncertainty by explicitly estimating an uncertainty score $\kappa_i$ for each candidate through an uncertainty quantification module. SAFER fine-tunes the model with awareness of candidates whose pseudo-annotations may lack reliability. The uncertainty score is integrated into a novel risk-aware loss function, mitigating the influence of uncertain optimal treatment labels.

## 3.2. Conformal Inference for FDR Control

In high-stakes domains such as treatment recommendation, it is vital to provide calibrated predictions and control the rate of incorrect decisions. To achieve this, we adopt a *conformal inference* procedure (Vovk et al., 1999; 2005; Shafer & Vovk, 2008) that selects a subset of plausible prediction candidates with a statistical coverage guarantee. We assume access to a *calibration set*

$$\mathcal{Z}_{\mathrm{cal}} = \left\{(\mathbf{x}_i, y_i)\right\}_{i=1}^n,$$

where each $\mathbf{x}_i$ is an input embedding and $y_i$ an observed treatment label. All pairs $(\mathbf{x}_i, y_i)$ are drawn i.i.d. from the same distribution as the training data set which used to train a predictor $f \colon \mathcal{X} \to \mathcal{Y}$. We then have $m$ new test samples $\{\mathbf{x}_{n+j}\}_{j=1}^m$ with true but *unobserved* labels $\{y_{n+j}\}_{j=1}^m$, each drawn i.i.d. from the same (unknown) data-generating process.

To address the risk of recommending incorrect treatments, we compute an *uncertainty score* $\kappa_i$ for each patient $i \in [n+m]$ via an "uncertainty map" module, and convert these scores into *conformal p-values* (e.g. Vovk et al., 1999; 2005; Bates et al., 2023; Jin & Candès, 2023; Liang et al., 2024) to control false discovery rate (FDR). The FDR is defined as

$$\mathrm{FDR} = \mathbb{E}\left[\frac{V}{R}\right], \text{where} \quad R = (\text{total \# of rejections}),$$
$$V = (\text{\# of false rejections}).$$

Concretely, we define a null hypothesis

$$H_j : \kappa_j \geq c, \ j = 1, \ldots, m, \tag{1}$$

for each test sample $j$, and reject (i.e., recommend) a subset $\mathcal{S} \subseteq \{1, \ldots, m\}$ of hypotheses while ensuring $\mathrm{FDR} \leq \alpha$, where $\alpha \in (0,1)$ is a user-specified tolerance, $c$ is a predefined uncertainty threshold.

This setup can be viewed as a standard *multiple testing* problem (Benjamini & Hochberg, 1995; 1997; Benjamini & Yekutieli, 2001; Efron, 2012) with $m$ null hypotheses $H_1, \ldots, H_m$. We compute a conformal p-value $p_j$ for each $H_j$ and apply an FDR-controlling procedure (e.g., Benjamini & Hochberg, 1995) to obtain a set $\mathcal{S}$ with the desired error rate $\alpha$. In binary classification tasks, the FDR serves as an analog to Type-I error control(Hastie, 2009). For regression problems with continuous responses, controlling errors is appropriate when each selected candidate incurs a comparable cost. This approach is particularly crucial in scenarios like medical decision-making and knowledge retrieval, where the cost of committing a type-I error can be significant and should be a primary consideration. In high-stakes clinical settings, a falsely recommended treatment can lead to adverse outcomes. By constraining $\mathrm{FDR} \leq \alpha$,

clinicians can trust that the expected fraction of incorrect recommendations remains safely bounded, thereby enhancing patient safety in the automated decision-making process.

## 4. Method

### 4.1. Dynamic Treatment Prediction

In practice, clinicians typically rely on both structured data and clinical notes to monitor disease progression and guide treatment decisions (Assale et al., 2019; Gangavarapu et al., 2020). Hence to mimic the real-world clinician decision-making process and recommend treatments at the next time step, we construct a unified time-series representation of the patient health embeddings by integrating these complementary data sources. To be more specific, given a patient $i$ represented by their multimodal electronic health record sequence $\mathbf{r}_i = \{(\mathbf{e}_i^1, \mathbf{o}_i^1), (\mathbf{e}_i^2, \mathbf{o}_i^2), \ldots, (\mathbf{e}_i^T, \mathbf{o}_i^T)\}$, where $\mathbf{e}_i^t \in \mathbb{R}^{d_\varepsilon}$ and $\mathbf{o}_i^t \in \mathbb{R}^{d_c}$ denote the structured EHR data and clinical notes at time $t$, our goal is to predict a treatment recommendation $\hat{y}_i^{T+1}$ for the next time step $T+1$.

**Inter-modality Temporal Dependency.** We first project the information from each data source into dense latent spaces. Specifically, we use BioClinicalBERT [2] to encode clinical notes, modeled as $\mathcal{X}^W$ (Alsentzer et al., 2019), which provides superior performance in encoding clinical text due to its bidirectional attention mechanism and domain-specific pretraining on large-scale biomedical and clinical corpora (Huang et al., 2023; Hu et al., 2024; Zhang et al., 2022). On the other hand, tabular data is encoded as $\mathcal{X}^C$ with normalization and one-hot encoding.

For a patient $p_i$, the sequence of each modality is aligned with the timestamps. The embedding layers $f \colon \mathcal{X}^A \to \mathbb{R}^{T \times d_k}$ are then applied to each modality, where $A \in \{E, O\}$ and $d_k$ is model dimensionality. To capture temporal dependencies within each modality, masked self-attention mechanisms are applied as,

$$\mathbf{S}_i^A = \mathrm{Softmax}\left(\frac{(\mathbf{X}_i^A \mathbf{W}_A^Q)(\mathbf{X}_i^A \mathbf{W}_A^K)^\top + \mathbf{M}}{\sqrt{d_k}}\right) \mathbf{X}_i^A \mathbf{W}_A^V + \mathbf{PE}. \tag{2}$$

where $\mathbf{M} \in \mathbb{R}^{T \times T}$ is the causal mask matrix, and $\mathbf{PE} \in \mathbb{R}^{T \times d_k}$ is the sinusoidal position encoding.

**Cross-modality information integration.** To effectively integrate information from different data sources, we design a cross-attention mechanism that enables different sequences to learn contextual information from each other, which can be formulated as,

$$\mathbf{H}_i = \left(\mathrm{softmax}\left(\frac{\mathbf{S}_i^O \mathbf{W}_E^Q (\mathbf{S}_i^E \mathbf{W}_E^K)^T}{\sqrt{d_k}}\right) \mathbf{S}_i^E \mathbf{W}_E^Q\right) \tag{3}$$

$$\oplus \left(\mathrm{softmax}\left(\frac{\mathbf{S}_i^E \mathbf{W}_O^Q (\mathbf{S}_i^E \mathbf{W}_O^K)^T}{\sqrt{d_k}}\right) \mathbf{S}_i^E \mathbf{W}_O^V\right). \tag{4}$$

---

[2]https://huggingface.co/emilyalsentzer/Bio_ClinicalBERT

where $\mathbf{S}_i^E$ and $\mathbf{S}_i^O$ are temporal-aware representations for each modality, and $\oplus$ denotes the concatenation operation. After integrating the static information embedding $\mathbf{x}_i^D$, the final representation is given by $\mathbf{h}_i = \mathbf{H}_i^T \oplus \mathbf{x}_i^D$, where $\mathbf{h}_i \in \mathbb{R}^{3d_k}$ represents the unified patient embeddings.

Subsequently, a feedforward network-based classification layer is applied to produce a probability distribution over medication classes for the next time step, as $f_\theta : \mathcal{H} \to \mathbb{R}^{|\mathcal{Y}|}$. The model is trained using cross-entropy loss to minimize prediction error across all instances.

### 4.2. Risk-Aware Fine-Tuning

After the dynamic treatment prediction module converges, we introduce a risk-aware fine-tuning procedure to account for label uncertainty as stated in Section 3.1, assuming reliable labels for surviving patients while uncertain labels for deceased patients. However, training exclusively on surviving patients significantly discards valuable information. To mitigate this, we treat labels for deceased patients as pseudo-labels and propose an uncertainty module to incorporate the brought risky information effectively.

**Uncertainty Estimation.** Here, we refine predictions for surviving patients by introducing a multilayer perceptron-based module $f_\phi$. This module takes the patient embeddings $h_i$, learned in the previous stage, as input to generate a new predictive distribution for each surviving patient. The uncertainty module is trained exclusively on surviving patients using cross-entropy loss to minimize the prediction error.

Since predictions have been refined for surviving patients, who exhibit more stable patterns and have reliable labels, the KL divergence between the logits before and after refinement can capture the distributional difference between surviving and deceased patients. This divergence, as shown in the equation below, can be interpreted as a measure of uncertainty for deceased patients.

During model inference, we quantify the predictive uncertainty by computing the KL divergence between the output distributions of the two modules $f_\theta$ and $f_\phi$ as follows,

$$\kappa_i = D_{\text{KL}}\left(p_\theta(\mathbf{h}_i) \,\|\, p_\phi(\mathbf{h}_i)\right) = \sum_{l=1}^{L} p_\theta(\widehat{y}_i = l|\mathbf{h}_i) \ln \frac{p_\theta(\widehat{y}_i = l|\mathbf{h}_i)}{p_\phi(\widehat{y}_i = l|\mathbf{h}_i)},$$ (5)

where $p(h_i) = \text{Softmax}(f(h_i))$ denotes the predicted probability distributions from both module, and $\widehat{y}_i$ represents the predicted class.

**Theorem 4.1.** *Let $h^- \sim P^-(h)$ and $h^+ \sim P^+(h)$ denote the latent representations of survivors and deceased patients respectively. Under the following conditions,*

*1. $D_{KL}(P^-(h) \,\|\, P^+(h)) > 0$, i.e. $P^- \neq P^+$*

*2. $f_\phi$ is L-Lipschitz continuous over latent representation space $\mathcal{H}$, where $h \in \mathcal{H}$,*

*that is there exists a constant $c > 0$, such that*

$$\mathbb{E}_{h \sim P^-}[\kappa_i] - \mathbb{E}_{h \sim P^+}[\kappa_i] \geq c > 0.$$ (6)

The proof details can be found in Appendix A.1. Hence in this regard, such KL divergence can capture the distributional difference between survival and deceased patients, served as a valid measure of uncertainty where $\kappa_i$ represents the prediction uncertainty.

*Remark 4.2.* A Lipschitz-constrained student (e.g. weight decay plus spectral normalisation) is a standard practice when one wishes to avoid uncontrolled extrapolation outside the training manifold. In our approach, we rely on the multilayer perceptron-based architecture, which inherently exhibits Lipschitz continuity when the activation functions are smooth and bounded, and the model's weights are appropriately regularized (Gouk et al., 2021).

**Risk-aware Loss Function.** The uncertainty term is incorporated into the loss function during fine-tuning. For surviving patients, uncertainty remains minimal, enabling the training process to prioritize these samples. Conversely, for deceased patients, the loss function penalizes significant deviations, ensuring the model effectively captures risk. The modified loss function is defined as follows,

$$\mathcal{L} = -\frac{1}{N} \sum_{i=1}^{N} (1 - \hat{\kappa}_i) \sum_{l=1}^{L} y_i \log p_\theta(\widehat{y}_i = l|h_i) + \gamma \kappa_i^2,$$ (7)

where $\hat{\kappa}_i$ is the normalized uncertainty term and $\gamma$ controls the regularization strength, penalizing overconfident predictions for high-risk cases.

## 5. Conformal Selection and FDR Control

We begin by fitting our risk-aware model on the training set. Subsequently, for calibration sample $\{(\mathbf{x}_i, y_i)\}_{i=1}^{n}$ and unlabeled test data $\{\mathbf{x}_{n+j}\}_{j=1}^{m}$, we compute the predicted uncertainty score $\widehat{\kappa}_i = \widehat{\kappa}(\mathbf{x}_i)$, for every $i \in [n+m]$. Here, $\widehat{\kappa} : \mathcal{X} \to \mathbb{R}$ is an uncertainty score predictor that depends on the patient health trajectory (e.g., time-series of clinical measurements), rather than on the observed treatment label $y_i$. As described in the previous section, $\kappa(\mathbf{x}_i)$ captures a "label mismatch risk" based on the distributional divergence between an *refined module* and a *fine-tuned module*. Crucially, this score does not require knowledge of the final treatment $y_i$. We require that $\widehat{\kappa}$ is computed in the same way for calibration and test samples. This consistent definition of $\kappa$ across calibration and test sets preserves the exchangeability necessary for valid conformal inference.

**Conformal p-Value.** Consider a test sample $j \in [m]$ for which we wish to test the hypothesis $H_j : \kappa_{n+j} \geq c$. Then

we define the conformal p-value

$$p_j = \frac{\sum_{i=1}^n \mathbb{1}\left\{\widehat{\kappa}_i < \widehat{\kappa}_{n+j}, \kappa_i \geq c\right\}}{n + 1}$$
$$+ \frac{U_j \cdot (1 + \sum_{i=1}^n \mathbb{1}\left\{\widehat{\kappa}_i = \widehat{\kappa}_{n+j}, \kappa_i \geq c\right\}))}{n + 1}. \quad (8)$$

where $U_j \sim \mathrm{Unif}(0, 1)$ are i.i.d. random variables used for tie-breaking under the multiple testing setting. Intuitively, $p_j$ measures how frequently the predicted calibration uncertainty scores $\{\widehat{\kappa}_i\}_{i=1}^n$ are less than or equal to the test uncertainty score $\widehat{\kappa}_{n+j}$, restricted to those $\kappa_i \geq c$.

Traditionally, conformal p-values are constructed to be *super-uniform* under the null, meaning that if the tested label (or score) truly matches the data-generating distribution, then $\mathbb{P}(p_j \leq \alpha) \leq \alpha$ for all $\alpha \in [0, 1]$ (Vovk et al., 2005; Lei et al., 2018). Here, the setup follows Jin & Candès (2023), who define $H_j : Y_{n+j} \leq c_j$ for random hypotheses based on unobserved labels. In the present formulation (8), the p-value satisfies a selective guarantee (Jin & Candès, 2023), namely:

$$\mathbb{P}\Big[\big(j \in \mathcal{S}\big) \wedge \big(p_j \leq \alpha\big)\Big] \leq \alpha, \quad \forall \alpha \in [0, 1], \quad (9)$$

where $\mathcal{S}$ is the final selected ("rejected") set. In other words, the joint event that $j$ is included in the recommendation set and $p_j \leq \alpha$ occurs with probability no larger than $\alpha$.

**Benjamini–Hochberg (BH) Procedure.** After computing conformal p-values $p_1, \ldots, p_m$ for the test samples, we control the FDR via the classic Benjamini–Hochberg (BH) algorithm (Benjamini & Hochberg, 1995). First, sort the p-values in ascending order:

$$p_{(1)} \leq p_{(2)} \leq \cdots \leq p_{(m)}.$$

Then, let

$$k = \max\left\{r : p_{(r)} \leq \frac{\alpha r}{m}\right\},$$

where $\alpha \in (0, 1)$ is the user-specified FDR threshold. If no such $r$ satisfies the inequality, we set $k = 0$. The BH procedure "rejects" the $k$ smallest p-values, i.e. the set $\{p_{(1)}, \ldots, p_{(k)}\}$. Accordingly, our *conformal selection* output is

$$\mathcal{S} = \left\{j \in [m] : p_j \leq p_{(k)}\right\},$$

meaning we only recommend labels whose p-values rank among these top $k$.

Below, we show that this procedure, using p-values of the form (8), controls the FDR under suitable assumptions.

**Theorem 5.1.** *Assume we have*

1. *The calibration data $\{(\mathbf{x}_i, y_i)\}_{i=1}^n$ and test data $\{\mathbf{x}_{n+j}\}_{j=1}^m$ are i.i.d., and data in $\{(\mathbf{x}_i, y_i)\}_{i=1}^n \cup \{\mathbf{x}_{n+l}\}_{l \neq j} \cup \{\mathbf{x}_{n+j}\}$ are mutually independent for any $j \in [m]$.*

2. *There exists some $M \geq 0$, such that $\sup_{\mathbf{x}} \kappa(\mathbf{x}) \leq M$.*

*Then, for any user-defined threshold $\alpha \in (0, 1)$, the BH-based conformal selection set $\mathcal{S}$ satisfies*

$$\mathrm{FDR} := \mathbb{E}\left[\frac{V}{\max\{1, R\}}\right]$$
$$= \mathbb{E}\left[\frac{\sum_{j=1}^m \mathbb{1}\{H_j \text{ is true}, j \in \mathcal{S}\}}{\max\left\{1, \sum_{j=1}^m \mathbb{1}\{j \in \mathcal{S}\}\right\}}\right] \leq \alpha. \quad (10)$$

The proof details are provided in Appendix A.2. By combining the conformal $p$-value construction in (8) with the BH procedure, our method ensures that the *expected proportion* of unreliable recommended treatments remains bounded by $\alpha$. By selecting an appropriate uncertainty score function $\kappa$, where higher $\kappa$ values correspond to lower plausibility of the candidate label $y$, we enable an intuitive calculation of conformal $p$-values. This framework provides a practical safety margin for high-stakes applications, while supporting flexible and data-driven selection of plausible labels.

# 6. Experiments

We empirically validate the SAFER model on two sepsis cohorts derived from publicly available datasets, Medical Information Mart for Intensive Care (MIMIC)-III covering over 40,000 ICU stays (2001–2012) (Johnson et al., 2016) and MIMIC-IV with over 65,000 ICU and 200,000 ED admissions (2008–2019) (Johnson et al., 2023a;b), to evaluate recommendation accuracy and FDR control. For this study, we define cohorts based on the sepsis-3 criteria (Singer et al., 2016), focusing on the early stages of sepsis management—24 hours prior to and 48 hours after sepsis onset. The treatment selection involves intravenous fluid and vasopressor dosage within a 4-hour window, mapped to a $5 \times 5$ medical intervention space, following Komorowski et al. (2018). Figure 2 shows the distribution of sepsis treatment co-occurrence in the two cohorts.

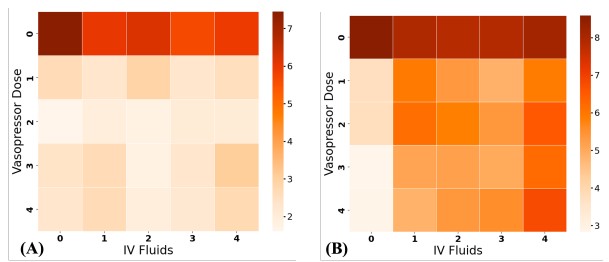

*Figure 2.* Comparative visualization of the treatment frequency matrix in $\log$ scale from two datasets. Panel (A) represents MIMIC-III, while Panel (B) corresponds to MIMIC-IV.

For each patient, we extract 5 types of static demographic variables and 44 types of time-series variables from the tabular data. We set the historical sequence length to 8. All

*Table 1.* The overall performance of SAFER and baseline methods. ($p < 0.05$)

| Methods | MIMIC-III | | | | | MIMIC-IV | | | | |
|---|---|---|---|---|---|---|---|---|---|---|
| | MI-AUC | MA-AUC | HR@3 | MRR@3 | ↓ Mortality | MI-AUC | MA-AUC | HR@3 | MRR@3 | ↓ Mortality |
| LSTM | 0.9122 | 0.7934 | 0.7481 | 0.8015 | 0.0915 | 0.9213 | 0.8121 | 0.7551 | 0.8066 | 0.1051 |
| RETAIN | 0.9257 | 0.8219 | 0.8324 | 0.8153 | 0.1994 | 0.9279 | 0.7851 | 0.8017 | 0.8052 | 0.1863 |
| TAHDNet | 0.9213 | 0.8017 | 0.7123 | 0.8109 | 0.2214 | 0.9157 | 0.8274 | 0.7554 | 0.8315 | 0.2466 |
| Naive RL | 0.7436 | 0.6025 | 0.5303 | 0.8891 | 0.0881 | 0.6782 | 0.5971 | 0.5068 | 0.8217 | 0.1172 |
| SRL-RNN | 0.8751 | 0.6215 | 0.7722 | 0.7916 | 0.3124 | 0.8781 | 0.6982 | 0.7824 | 0.8151 | 0.3219 |
| ACIL | 0.8219 | 0.7012 | 0.8013 | 0.8313 | 0.3212 | 0.8854 | 0.7135 | 0.8319 | 0.8441 | 0.3782 |
| ISL | 0.8903 | 0.7785 | 0.7623 | 0.7521 | 0.2783 | 0.8713 | 0.7315 | 0.7741 | 0.7229 | 0.3118 |
| **SAFER** | **0.9407** | **0.8672** | **0.8517** | **0.9017** | **0.3891** | **0.9356** | **0.8755** | **0. 8713** | **0.8698** | **0.4562** |

clinical notes were aligned to the closest timestamp. To handle outliers, we applied the interquartile range (IQR) method for removal and imputed missing values using the $k$-nearest neighbors approach. Subsequently, all variables were rescaled to the [0,1] interval using z-score normalization. The two datasets were randomly split into training, calibration (validation), and test sets in an 80%/10%/10% ratio via patient-level splits to ensure no patient overlap, under the assumption that the entire dataset is i.i.d. sampled from a common distribution.

The uncertainty score $\kappa_i$ quantifies the model's confidence in its treatment predictions. Once the uncertainty scores are computed for the training set, we train the uncertainty score predictor $\widehat{\kappa}$ on the calibration and test sets using standard machine learning models leveraging the full embedding feature space $\mathcal{X}$. Model performance is assessed by evaluating the average FDR across 500 independent experiments. Appendix C.3 provides a sensitivity analysis of several hyperparameters, including the length of historical information, hidden dimension, and $\gamma$ in the loss function.

### 6.1. Evaluation Metrics

To evaluate the performance of SAFER and other baselines, we report MRR@3 and HR@3 for treatment ranking, as well as Micro AUC and Macro AUC for assessing predictive performance in the multiclass classification setting of DTR. Additionally, we report the counterfactual mortality rate reduction, which is a measure of how recommended treatments might have improved survival outcomes relative to real-world clinical actions (Laine et al., 2020; Kusner et al., 2017; Valeri et al., 2016), to validate the effectiveness of the recommended treatment as part of an offline value estimation. The details for valid counterfactual mortality rate calculation are provided in Appendix B.

### 6.2. Baseline Methods

For validating the effectiveness of SAFER, we selected several baseline methods for comparison. The baselines can be categorized as sequential embedding based and reinforcement learning based approaches.

Sequential embedding methods include: **LSTM** (Hochreiter & Schmidhuber, 1997), widely used time-series prediction model; **RETAIN** (Choi et al., 2016), a two-level neural attention-based model that highlights key visit sequences for treatment prediction; **TAHDNet** (Su et al., 2022), a hierarchical temporal dependency network for dynamic treatment prediction. Reinforcement learning based methods include **Naive Baseline for RL**(Luo et al., 2024), a simple rule-based approach for benchmarking RL algorithms. **SRL-RNN** (Wang et al., 2018), which integrates supervised learning with RL using survival signals as rewards. **ACIL** (Wang et al., 2020), an adversarial imitation learning approach that optimizes treatment by learning from both successful and failed trajectories. **ISL** (Jiang et al., 2023), a prototype-based model ensuring treatment actions align with learned representations. For fair comparison, all methods use the same data sources. Methods lacking native text processing capabilities incorporate BioClinicalBERT embeddings for clinical notes. Given our assumption that only surviving patients have fully reliable labels, we conduct primary evaluations on this subset, while assessing generalization to deceased patients through counterfactual mortality rate analysis across the entire dataset population.

### 6.3. Overall Performance

Table 1 shows the overall performance of our proposed SAFER and baseline methods. Our analysis reveals several key observations as follows.

First, RL methods demonstrate consistent underperformance on classification benchmarks in general, especially with the macro-AUC metric on MIMIC-III revealing a 16.6% deficit compared to sequence-based counterpart baselines on average. This probably stems from severe class imbalance of treatment label (Class 0 takes 61.2% percent), where sparse disease-specific reward signals prove insufficient for distinguishing between classes. Consequently, RL agents fail to develop discriminative policies, resulting even worse performance compared with simple baselines in mortality prediction. While imitation learning approaches partially mitigate this issue, their performance remains suboptimal

*Table 2.* The performance of SAFER and its variants. ($p < 0.05$)

| Variants | MIMIC-III | | | | | MIMIC-IV | | | | |
|---|---|---|---|---|---|---|---|---|---|---|
| | MI-AUC | MA-AUC | HR@3 | MRR@3 | ↓ Mortality | MI-AUC | MA-AUC | HR@3 | MRR@3 | ↓ Mortality |
| SAFER-*F* | 0.9059 | 0.7140 | 0.7254 | 0.8067 | 0.2402 | 0.8853 | 0.6851 | 0.7199 | 0.7542 | 0.2315 |
| SAFER-*N* | 0.8655 | 0.7651 | 0.7523 | 0.7803 | 0.2951 | 0.8897 | 0.7841 | 0.7553 | 0.8029 | 0.3875 |
| SAFER-*U* | 0.9188 | 0.8237 | 0.8321 | 0.8769 | 0.2982 | 0.9231 | 0.8317 | 0.8451 | 0.8544 | 0.3765 |
| SAFER | **0.9407** | **0.8672** | **0.8517** | **0.9017** | **0.3891** | **0.9356** | **0.8755** | **0.8713** | **0.8698** | **0.4562** |

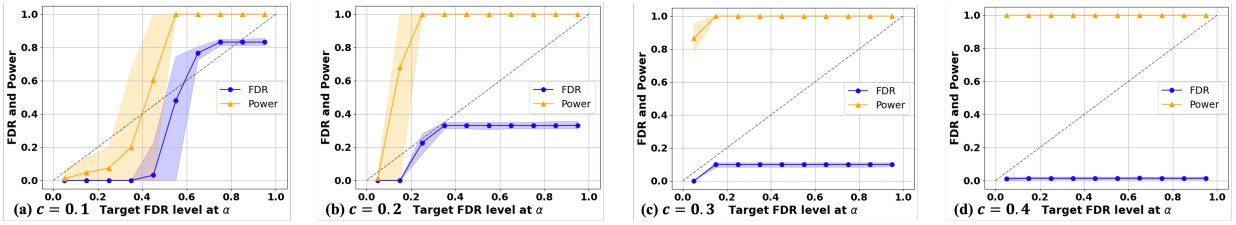

*Figure 3.* FDR and power curves across different target $\alpha$ level and varies uncertainty threshold $c$ on MIMIC-III with Ridge Regression.

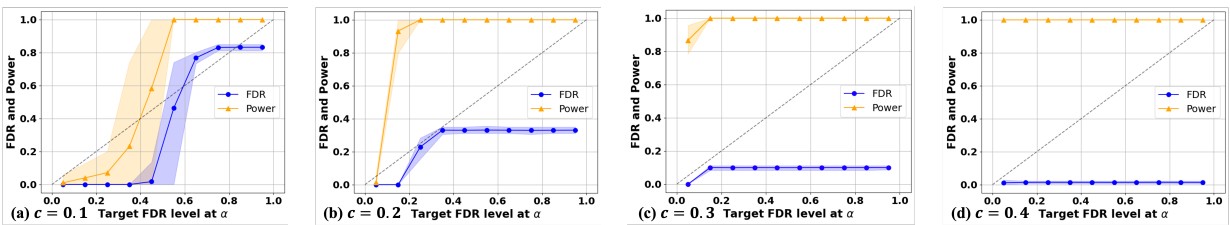

*Figure 4.* FDR and power curves across different target $\alpha$ level and varies uncertainty threshold $c$ on MIMIC-IV with Ridge Regression.

due to high patient heterogeneity in treatment responses.

Second, while sequence embedding-based methods excel in classification tasks, their effectiveness declines in ranking metrics compared to RL approaches. This is probably because embedding methods, while capturing global patterns, fail to distinguish fine-grained treatment efficacy differences. RL methods, by contrast, explicitly optimize treatment policies using reward signals tied to cure outcomes, enabling differentiated pattern learning across survival and death trajectories. Traditional embedding approaches indiscriminately encode all treatment sequences, introducing uncertainty due to distribution shifts between surviving and deceased patients, ultimately reducing performance in fine-grained ranking. However, our SAFER model, though embedding-based, effectively mitigates this issue.

Finally, our proposed method demonstrates superior performance across both classification and ranking metrics by effectively modeling EHR data sequences across multiple modalities and integrating label uncertainty from deceased patients. This results in more robust and generalizable predictions. Beyond these metrics, we further analyze the impact of our model on counterfactual mortality rate reduction. Overall, sequential embedding-based methods tend to un-

derperform compared to RL methods in this aspect, as they indiscriminately encode all treatment sequences rather than learning distinct treatment strategies tailored to different patient outcomes. However, our risk-aware fine-tuning module significantly mitigates this limitation by incorporating uncertainty information and applying a penalty mechanism to focus training on reliable labels. As a result, SAFER achieves a substantial reduction in mortality rate, showcasing the effectiveness of our approach in recommending treatments with valuable real-world meanings.

### 6.4. FDR Control

**SAFER strictly controls the FDR.** Figure 3 and Figure 4 present the realized FDR and power curves for SAFER on MIMIC-III & IV respectively at various target FDR levels $\alpha \in \{0.05, 0.10, \dots, 0.95\}$ across different uncertainty thresholds $c$. Here, power is defined as $\mathbb{E}\left[\frac{\sum_{j=1}^{m} \mathbb{1}\{\hat{H}_j \text{ is false}, j \in \mathcal{S}\}}{\max\left\{1, \sum_{j=1}^{m} \mathbb{1}\{H_j \text{ is false}\}\right\}}\right]$. The uncertainty score predictor $\hat{\kappa}$ is trained with all feature embeddings via Ridge Regression. Results trained from additional regression predictors are provided in Appendix C.2. The results show that SAFER maintains strict control over the FDR at the specified level $\alpha$, which stabilizes as $\alpha$ increases. Also, the power

curve asymptotically converges to one with increasing $\alpha$, indicating that SAFER selects all confident candidates without exceeding the FDR constraint.

**Choice of Uncertainty Score Threshold.** The performance of SAFER depends on the choice of the uncertainty threshold $c$, evaluated over $c \in \{0.1, 0.2, 0.3, 0.4\}$. As $c$ increases, both FDR and power stabilize more quickly, converging to a constant value and one respectively. For $c = 0.1$, the FDR curve remains steady, and power remains close to zero until $\alpha = 0.7$. In contrast, the FDR reaches a constant value of 0.1 even at $\alpha = 0.1$ with power as high as 1. SAFER employs KL-divergence to quantify model prediction uncertainty while there is no universally accepted threshold to determine when two distributions differ meaningfully. Therefore, as illustrated in Figure 3 and Figure 4, we provide a practical guideline for selecting the uncertainty score threshold in real-world applications.

### 6.5. Ablation Study

We conduct ablation studies to validate the contribution of key components in SAFER, as demonstrated in Table 2.

The first variant, **SAFER-*F***, removes the risk-aware fine-tuning process, directly using the initial prediction module for inference. This leads to notable performance degradation, especially in Macro-AUC and counterfactual mortality rate on MIMIC-IV, which has a high proportion of deceased patients. This confirms our hypothesis that accounting forlabel uncertainty is crucial for robust decision-making.

For the second variant **SAFER-*N***, we remove clinical notes, but relying only on structured EHR data. Performance deteriorates significantly across all metrics, demonstrating that structured data alone fails to capture essential contextual cues from clinical narratives, highlighting the importance of textual information in modeling temporal dependencies.

Finally, **SAFER-*U*** removes the fine-tuning step and mitigates uncertainty by training only on surviving patients. Even within the survival subset, its performance remains inferior to SAFER, with a marked decline in counterfactual mortality rate. These findings emphasize the importance of incorporating deceased patients in training and validate our approach to handling the uncertainty they introduce.

### 7. Conclusion

We have introduced SAFER, an end-to-end **multimodal** DTR framework that delivers **reliable treatment recommendations** with uncertainty quantification and theoretical guarantees. Compared with existing DTR frameworks, we provide a solution that may be more suitable to high-stakes scenarios, ensuring safer and more trustworthy decision-making. It outperforms SOTA baselines across multiple recommendation metrics while achieving the greatest reduc-

tion in mortality rates. These results underscore SAFER's potential for trustworthy and risk-aware decision support in real-world clinical settings.

While this work primarily addresses inherent label uncertainty, real-world clinical data present broader challenges, including missing labels, latent confounders, and comorbidities. Tackling these complexities is essential for developing more generalizable and clinically grounded DTR frameworks. Future research can also build upon our approach to alternative error control notions beyond FDR, further improving the robustness and safety of treatment recommendations.

### Acknowledgements

This work was partially supported by the National Institutes of Health (NIH) under grant numbers U01TR003709, U24MH136069, RF1AG077820, R01AG073435, R56AG074604, R01LM013519, R01LM014344, R01DK128237, R21AI167418, and R21EY034179, and by the National Science Foundation (NSF) under grant numbers IIS-2006387 and IIS-2040799.

### Impact Statement

Dynamic treatment regimes (DTRs) play a crucial role in precision medicine by enabling personalized and adaptive treatment plans that have the potential to significantly improve patient outcomes. Although a wide range of DTR approaches show great promise through tailored treatment strategies based on patient responses, their application in high-stakes clinical settings necessitates rigorous and responsible implementation.

This work emphasizes the critical responsibility of the research community to ensure safety, ethical standards, and tangible benefits to patient care when advancing such technologies. SAFER addresses the inherent unreliability in clinical data by incorporating uncertainty quantification and mitigating prediction uncertainty, all while providing statistical guarantees. By controlling the false discovery rate in treatment recommendations, our approach safeguards patient trust and ensures that advancements in DTR do not come at the expense of patient safety or ethical integrity.

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

# A. Technical Proofs

## A.1. Lower-Bound Proof for Theorem 4.1.

*Restate of Theorem 4.1.* We have two latent-representation distributions $P^-(h)$ and $P^+(h)$ over the same latent space $\mathcal{H}$, corresponding to *deceased* and *surviving* patients, respectively. Let $h^- \sim P^-(h)$ and $h^+ \sim P^+(h)$. We define the *predictive uncertainty* at $h$ as

$$\kappa(h) = D_{\mathrm{KL}}\Big(p_\theta(y \mid h) \,\Big\|\, p_\phi(y \mid h)\Big),$$

where $p_\theta$ is the "teacher" module's distribution and $p_\phi$ is the "student" module's distribution (trained with risk-aware fine-tuning). The theorem claims that under:

1. $D_{\mathrm{KL}}\big(P^-(h) \,\|\, P^+(h)\big) > 0$, i.e. $P^- \neq P^+$,

2. $f_\phi$ is $L$-Lipschitz continuous over $\mathcal{H}$ (so $p_\phi(y \mid h)$ cannot *sharply* change as $h$ varies),

we have

$$\mathbb{E}_{h \sim P^-}[\kappa(h)] \;>\; \mathbb{E}_{h \sim P^+}[\kappa(h)] \quad \text{with a strictly positive lower bound.}$$

*Proof of Theorem 4.1.* Since $D_{\mathrm{KL}}\big(P^- \,\|\, P^+\big) > 0$, there must be at least one measurable subset $\mathcal{G} \subseteq \mathcal{H}$ on which $P^-$ places strictly greater mass than $P^+$ (or vice versa). Concretely, there exists $\epsilon > 0$ such that

$$P^-(\mathcal{G}) - P^+(\mathcal{G}) \;\geq\; \epsilon.$$

Since if no such region existed, $P^-$ would equal $P^+$ almost everywhere, contradicting $D_{\mathrm{KL}}(P^- \| P^+) > 0$.

Also as we have

$$\kappa(h) = D_{\mathrm{KL}}\Big(p_\theta(y \mid h) \,\|\, p_\phi(y \mid h)\Big).$$

If $p_\phi$ is $L$-Lipschitz in $h$, then as $h$ varies within a small neighborhood, the entire predicted distribution $p_\phi(y \mid h)$ cannot drastically jump to match $p_\theta(y \mid h)$ perfectly, unless the underlying latent distributions $P^-, P^+$ are aligned. Since $P^- \neq P^+$, there is a region $\mathcal{G}$ in latent space where $p_\phi$ cannot "annihilate" the mismatch in $p_\theta$. That is on a measurable set $\mathcal{G} \subset \mathcal{H}$, the teacher predictions differ from the student by at least a fixed amount:

$$\|p_\theta(\cdot \mid h) - p_\phi(\cdot \mid h)\|_1 \geq \delta \quad \text{for all } h \in \mathcal{G}, \tag{2}$$

with constants $\delta > 0$. Hence, $\kappa(h)$ is bounded away from zero on some portion of $\mathcal{G}$ with nontrivial measure under $P^-$,

$$\mathbb{E}_{h \sim P^-}[\kappa(h)] = \int_{\mathcal{H}} \kappa(h)\, dP^-(h) \;\geq\; \int_{\mathcal{G}} \kappa(h)\, dP^-(h) \geq \frac{1}{2}\delta^2 P^-(\mathcal{G}). \tag{6}$$

Where the second inequality follows from Pinsker's inequality which stated in many standard results (*e.g.*, (Sriperumbudur et al., 2009; Tsybakov & Tsybakov, 2009)).

Split the survivor expectation into the same region $\mathcal{G}$ and its complement:

$$\mathbb{E}_{h \sim P^+}[\kappa(h)] = \underbrace{\int_{\mathcal{G}} \kappa(h)\, dP^+(h)}_{(a)} + \underbrace{\int_{\mathcal{H} \setminus \mathcal{G}} \kappa(h)\, dP^+(h)}_{(b)}.$$

For the first term (a), we can control $\kappa$ by Lipschitzness. Fix $h \in \mathcal{G}$ and pick $h'$ with $P^+$-density such that $d(h, h') \leq r$ for a radius $r > 0$ (possible because $\mathrm{supp}(P^+) = \mathcal{H}$ in practice). Applying assumption 1 and again Pinsker's inequality,

$$\kappa(h) \;\leq\; D_{\mathrm{KL}}\left(p_\theta \,\|\, p_\phi(\cdot \mid h')\right) + CL^2 r^2,$$

where $C$ is an absolute constant. Choosing $r$ small makes this term negligible compared with $\delta$. Denote the resulting bound by $\varepsilon_1$. For the second term (b), as the global survivor risk can be tuned at most $\varepsilon$, therefore (b) $\leq \varepsilon$.

Collecting the two parts we have

$$\mathbb{E}_{h\sim P^+}[\kappa(h)] \ \leq \ \varepsilon + \varepsilon_1.$$

Subtract this from the lower bound of $\mathbb{E}_{h\sim P^-}[\kappa(h)]$:

$$\mathbb{E}_{h\sim P^-}[\kappa(h)] - \mathbb{E}_{h\sim P^+}[\kappa(h)] \ \geq \ \frac{1}{2}\delta^2 P^-(\mathcal{G}) - (\varepsilon + \varepsilon_1)$$

$$= \ \underbrace{\left(\frac{1}{2}\delta^2\pi - (\varepsilon + \varepsilon_1)\right)}_{=:c}.$$

Where $\pi := P^-(\mathcal{G}) - P^+(\mathcal{G}) > 0$ from the first assumption and $\delta > 0$, we can easily pick training and regularisation so that $\varepsilon, \varepsilon_1$ are small enough, hence $c > 0$.

Putting it together,

$$\mathbb{E}_{h\sim P^-}[\kappa(h)] - \mathbb{E}_{h\sim P^+}[\kappa(h)] \ \geq \ c > 0$$

with an explicit constant $c = \frac{1}{2}\delta^2\pi - \varepsilon - \varepsilon_1$. This establishes that deceased-patient latents (drawn from $P^-$) systematically lead to higher KL-based uncertainty $\kappa(\cdot)$ than do survivor latents from $P^+$, giving a strict positive gap on average. Hence the theorem's statement follows.

$\square$

*Remark* A.1. By forcing $p_\phi$ to remain continuous with respect to $h$, we guarantee that if latent embeddings of deceased patients differ significantly from those of survivors, the student's predicted distributions cannot "collapse" to match the teacher's everywhere in $\mathcal{H}$. Hence, the *KL uncertainty* $\kappa(h)$ for $h^- \sim P^-$ stays measurably larger on average than for $h^+ \sim P^+$, ensuring $\mathbb{E}_{P^-}[\kappa] > \mathbb{E}_{P^+}[\kappa]$ by at least a positive margin.

*Remark* A.2. $\delta$ can be estimated on a validation split by the empirical minimum teacher–student $\ell_1$ gap over high-mortality clusters; $\pi$ follows from any two-sample test on the latent representations; $\varepsilon$ is the held-out risk of the student model on survivors.

*Remark* A.3. Tighter bounds. One may replace Pinsker's inequality by the Bretagnolle–Huber inequality or by Csiszár–Kullback–Pinsker to sharpen $c$. The qualitative conclusion of strict positivity remains unchanged.

*Remark* A.4. If the student is *Bayes-optimal* for $P^+$ (in the limit of infinite positive data) and the teacher is Bayes-optimal for the mixture,then $\mathbb{E}_{P^+}[\kappa] = 0$ exactly, and the proof simplifies to analysing $P^-$ only, yielding $c = \frac{1}{2}\delta^2 P^-(\mathcal{G})$.

## A.2. FDR Control Proof of Theorem 5.1

*Restate of Theorem 5.1.* Let $\kappa : \mathcal{X} \to \mathbb{R}$ be an uncertainty function, and $\sup_x \kappa(x) \leq M$ for some $M \geq 0$. We have $n$ i.i.d. calibration samples $\{(\mathbf{x}_i, y_i)\}_{i=1}^n$ and $m$ i.i.d. test inputs $\{\mathbf{x}_{n+j}\}_{j=1}^m$, all mutually independent in the sense that *any* subset excluding index $j$ is jointly independent of the data at index $j$. For each test point $j$ we define a null hypothesis

$$H_j : \ \kappa_{n+j} \ \geq \ c,$$

and the conformal p-value $p_j$ as in (8), namely

$$p_j = \frac{\sum_{i=1}^n \mathbb{1}\left\{\widehat{\kappa}_i < \widehat{\kappa}_{n+j},\ \kappa_i \geq c\right\} + 1}{n + 1} + \frac{U_j \cdot \left(1 + \sum_{i=1}^n \mathbb{1}\left\{\widehat{\kappa}_i = \widehat{\kappa}_{n+j},\ \kappa_i \geq c\right\}\right)}{n + 1},$$

where $U_j \sim \text{Unif}(0,1)$ are i.i.d. tie-breaking variables. Let the *Benjamini–Hochberg (BH)* procedure at level $\alpha \in (0,1)$ be applied to $\{p_j\}_{j=1}^m$, producing a selection (rejection) set $\mathcal{S}$. Denote $R = |\mathcal{S}|$ and

$$V \ = \ \sum_{j=1}^m \mathbb{1}\left\{j \in \mathcal{S}, H_j \text{ is true}\right\},$$

the number of false rejections. Then under the above assumptions, the false discovery rate (FDR) is

$$\text{FDR} \ = \ \mathbb{E}\left[\frac{V}{\max\{1, R\}}\right] \leq \alpha.$$

*Proof of Theorem 5.1.* We define the *nonconformity score* $J$ as

$$J(\mathbf{x}, y) = \kappa(\mathbf{x}) + 2M \cdot \mathbb{1}\{ y \geq c \}.$$

Thus, if $y < c$ (i.e., if the label is "below" the critical threshold $c$), $J(\mathbf{x}, y) = \kappa(x)$. On the other hand, if $y \geq c$, then $J(\mathbf{x}, y) = \kappa(\mathbf{x}) + 2M$. The nonconformity score $J(\mathbf{x}, y)$ preserves the monotonicity property in terms of $y$. Thus if we define $J_i = J(\mathbf{x}_i, y_i)$, $\widehat{J}_i = J(\mathbf{x}_i, c)$ for every $i \in [n + m]$, the conformal p-value defined in Equation (8) converts to

$$p_j = \frac{\sum_{i=1}^{n} \mathbb{1}\left\{ J_i < \widehat{J}_{n+j} \right\}}{n + 1} + \frac{U_j \cdot (1 + \sum_{i=1}^{n} \mathbb{1}\left\{ J_i = \widehat{J}_{n+j} \right\})}{n + 1},$$

as defined in Jin & Candès (2023). To ensure the completeness of the proof, we adapt major proof procedures from Theorem 3 in Jin & Candès (2023) as following.

Using $J(\cdot, \cdot)$ in a standard conformal scheme (Vovk et al., 2005; Lei et al., 2018), we obtain p-values $p_1, \ldots, p_m$ of the Equation (8) (including $U_j$ for tie-breaking). By *exchangeability* of calibration and test data, plus the monotonicity of $J$, each $p_j$ is "selectively super-uniform" with respect to its null $H_j$ (Jin & Candès, 2023); that is, for every $\alpha \in [0, 1]$,

$$\mathbb{P}\Big[(j \in \mathcal{S}) \wedge (p_j \leq \alpha)\Big] \leq \alpha.$$

Roughly, this property ensures that $p_j$ behaves conservatively if $H_j$ is true. Moreover, one typically invokes a PRDS condition (Positive Regression Dependence on a Subset) or mutual independence across $\{p_j\}_{j=1}^{m}$ to ensure the BH procedure can be applied with classical guarantees (Benjamini & Yekutieli, 2001; Efron, 2012).

Then under i.i.d. sampling $\{(\mathbf{x}_i, y_i)\}_{i=1}^{n}$ plus $\{\mathbf{x}_{n+j}\}_{j=1}^{m}$ and monotonic $J$, the random variables $p_1, \ldots, p_m$ exhibit either independence or positive correlation that meets PRDS assumptions (see, e.g., Bates et al., 2023; Jin & Candès, 2023). Hence, each true null label $j$ effectively satisfies $p_j \sim$ selective super-uniform with respect to $H_j$.

Finally, we show FDR $\leq \alpha$ once BH is applied to the p-values $(p_1, \ldots, p_m)$ at level $\alpha$. Let $\mathcal{S} = \{ j : p_j \leq p_{(k)} \}$ denote the BH rejection set, where

$$k = \max\left\{ r : p_{(r)} \leq \frac{\alpha r}{m} \right\}.$$

Define indicator random variables $R_j = \mathbb{1}\{j \in \mathcal{S}\}$ and $T_j = \mathbb{1}\{H_j \text{ is true}\}$. Then

$$\text{FDR} = \mathbb{E}\Big[\frac{\sum_{j=1}^{m} T_j R_j}{\max\{1, \sum_{j=1}^{m} R_j\}}\Big] = \mathbb{E}\Big[\frac{1}{\max\{1, \sum_j R_j\}} \sum_{j=1}^{m} T_j R_j\Big].$$

From the property of BH under PRDS super-uniform p-values (Benjamini & Hochberg, 1995; Benjamini & Yekutieli, 2001; Bates et al., 2023; Jin & Candès, 2023), we have

$$\mathbb{E}\big[T_j R_j\big] \leq \alpha \mathbb{E}\big[R_j\big],$$

summing over $j$ and employing the usual BH bounding technique, it follows that

$$\text{FDR} = \mathbb{E}\Big[\frac{\sum_{j=1}^{m} T_j R_j}{\max\{1, \sum_j R_j\}}\Big] \leq \alpha.$$

In short, the fraction of wrongly rejected true nulls among all rejections remains at or below $\alpha$ in expectation. This completes the proof.

$\square$

## B. Counterfactual Mortality Calculation

Suppose $M_i$ represents the mortality event, $M_i(y)$ refers to the potential outcome under the treatment arm $Y_i = y$, and $\mathbf{X}_i \in \mathbb{R}^{T \times d_k}$ is the measured counfounders for the $i$-th patient. To estimate the effectiveness of our recommended treatment plans, we evaluate the model with the reduction in counterfactual mortality rate, we train an additional LSTM-based neural network as a counterfactual mortality prediction model. During training, this model takes patient features and ground-truth

treatments as input and is optimized using Binary Cross Entropy (BCE) loss to predict the probability of death as a binary classification task. During inference, the trained model estimates a counterfactual mortality rate by applying the model to patient features combined with the recommended treatment (i.e., the treatment predicted by our DTR model). The decrease in mortality rate is then defined as the difference between this estimated counterfactual mortality and the actual observed mortality rate under standard clinical practice. This analysis is based on the following assumptions regarding potential outcomes:

(B1) **No interference.** $M_i(Y_i)$ depends only on $Y_i$.

(B2) **No hidden variability.** Each unit has unique $M_i(y)$.

(B3) **Ignorability / No Unmeasured Confounding.** Conditioned on the measured covariates $\mathbf{X}_i$, the potential outcomes $M_i(y)$ are independent of the assigned treatment. That is,

$$\big\{M_i(y)\big\}_{y\in\mathcal{Y}} \;\perp\; Y_i \;\mid\; \mathbf{X}_i.$$

(B4) **Positivity (Overlap).** Every treatment arm $y$ has a nonzero probability of being assigned given $\mathbf{X}_i$, so $P(Y_i = y \mid \mathbf{X}_i) > 0$ for all $y \in \mathcal{Y}$.

We maintain the classic stable unit treatment value (SUTVA) assumption (Rubin, 1980) that no interference between units in (B1) and no hidden variations of treatments occur in (B2), If patient $i$ actually receives treatment $y$, then the observed mortality $M_i$ coincides with the potential outcome $M_i(y)$, which allows us to assume that $M = \sum_y Y M(y)$ almost surely. (B3) is a standard assumption on the ignorability of treatment assignment (Shi et al., 2023). Together, these assumptions allow us to view $M_i(y)$ as a well-defined counterfactual, enabling estimation of counterfactual mortality under different model-predicted treatments. Under these assumptions, we have $\mu_y^\star(\mathbf{X}) = \mathbb{E}\{M \mid Y = y, \mathbf{X}\}$, where $\mu_y^\star(\mathbf{X}) = \mathbb{E}\{M(y) \mid \mathbf{X}\}$.

# C. Experiment Details

## C.1. Dataset Details

**Dataset Statistical** Table 3 shows the statistical details of the two cohort datasets we use.

| Dataset | #Survival | #Deceased | #Avg Len | #Avg Notes |
|---------|-----------|-----------|----------|------------|
| MIMIC-III | 3118 | 427 | 11.75 | 40.65 |
| MIMIC-IV | 19450 | 3786 | 11.50 | 29.08 |

*Table 3.* The dataset statistics.

**Attribute** Table 4 presents the attributes used in our experiments.

| Attribute Type | Attribute Name |
|----------------|----------------|
| Demographics | Gender, Age, Re_admission, Weight_kg, Height_cm |
| Vital Signals | GCS, RASS, HR, SysBP, MeanBP, DiaBP, RR, Temp_C, CVP, PAPsys, PAPmean, PAPdia, CI, SVR, FiO2_1, O2flow, PEEP, TidalVolume, MinuteVentil, PAWmean, PAWpeak, PAWplateau, Potassium, Sodium, Chloride, Glucose, BUN, Creatinine, Magnesium, Calcium, , SGOT, SGPT, Total_bili, Direct_bili, Total_protein, Albumin, Troponin, CRP, Hb, Ht, RBC_count, WBC_count, Platelets_count, PTT, PT, ACT, INR, Arterial_pH, paO2, paCO2, Arterial_BE, Arterial_lactate, HCO3, ETCO2, SvO2, mechvent, extubated, Shock_Index, PaO2_FiO2, SOFA, SIRS |

*Table 4.* The attribute used in the experiments.

**The label frequency on suvivor subset.** Since we evaluate most metrics on the survivor subset, we also report the label frequency on the survivor subset as Figure 5 shows. In our experiments, we follow established protocols from prior sepsis treatment studies (e.g., Komorowski et al., 2018) by discretizing the intravenous fluid and vasopressor dosages into 5 bins each. Specifically, any absence of medication constitutes the zero bin, while the remaining dosages are partitioned into four additional bins according to empirical quantiles. This results in a $5 \times 5$ grid, forming 25 discrete treatment classes, where each class corresponds to a unique combination of fluid and vasopressor dosage levels (i.e., (fluid bin) × (vasopressor bin)). The distribution of these treatment classes are visualized in Figure 2 of the manuscript.

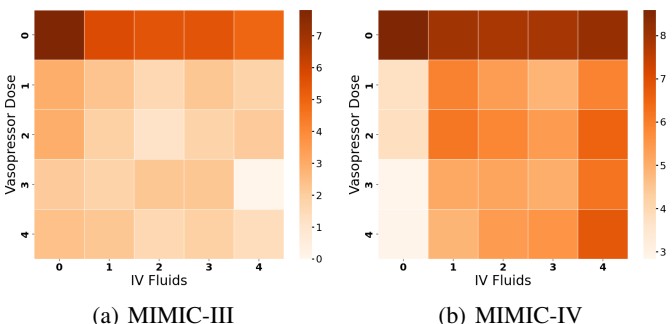

(a) MIMIC-III      (b) MIMIC-IV

*Figure 5.* Comparative visualization of the treatment frequency matrix in $\log$ scale from the survivor subset of two datasets. Panel (A) represents MIMIC-III, while Panel (B) corresponds to MIMIC-IV.

## C.2. FDR Control

In this section, we present the FDR control results using Linear Regression on the MIMIC-III dataset (Figure 6) and the MIMIC-IV dataset (Figures 7), further validating the effectiveness of our FDR control mechanism.

## C.3. Parameter Sensitivity

In this section, we discuss the sensitivity of SAFER to on three key hyperparameters, the historical information sequence length $L$, the hidden dimensionality $d_h$ and the risk regularization coefficient $\gamma$ in the loss function. We report Macro-AUC

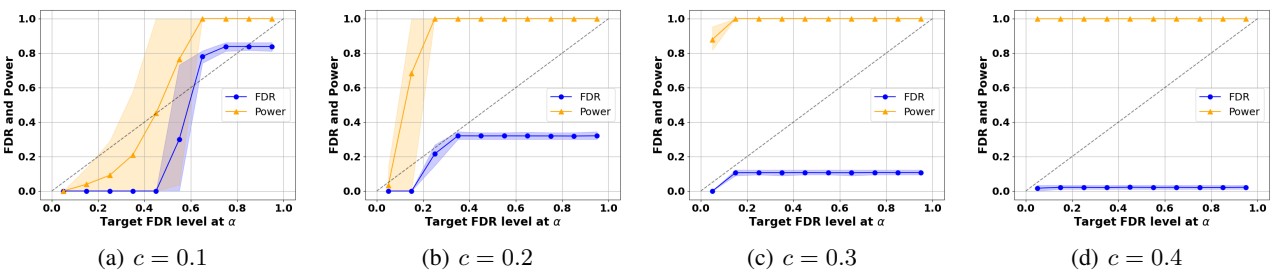

Figure 6. FDR and power curves across different target $\alpha$ level with Linear Regression on MIMIC-III.

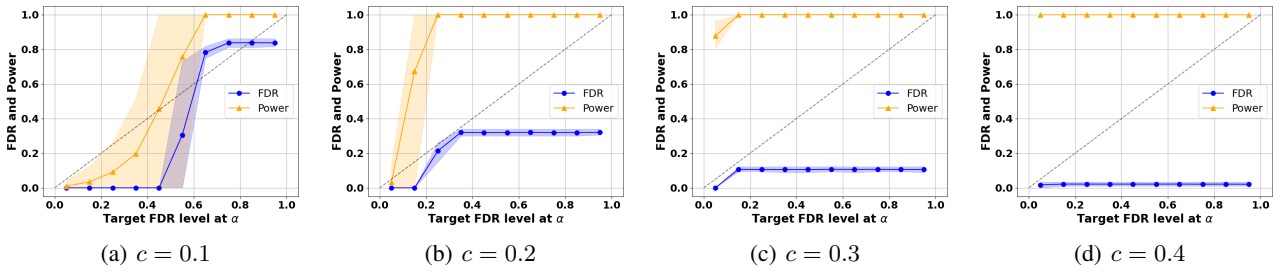

Figure 7. FDR and power curves across different target $\alpha$ level with Linear Regression on MIMIC-IV.

and the reduction in counterfactual mortality rate, two core evaluation metrics that reflect recommendation accuracy and treatment effectiveness.

To ensure fair comparison and robustness, we perform a grid search over a predefined range of values for each parameter while holding others fixed. The selected values correspond to those that jointly optimize both performance metrics on the validation set.

**Historical sequence length** $L$: Figure 8 illustrates that SAFER's performance improves significantly as the sequence length increases initially, before stabilizing at a consistent level. This trend likely occurs because shorter sequences lack sufficient information for accurate predictions. To ensure a fair comparison across datasets, we set the sequence length to 8 for all experiments.

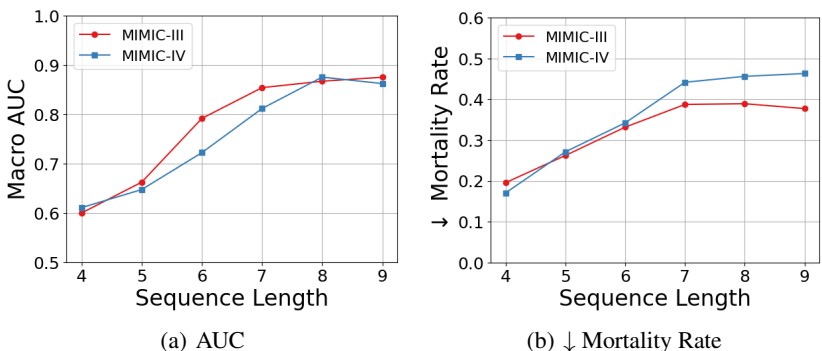

Figure 8. The performance of SAFER under different historical information sequence length $L$

**Hidden dimensionality** $h_d$: Figure 9 illustrates the performance of SAFER across different hidden dimensionalities $h_d$, showing that both low and excessively high dimensions degrade model performance. A lower-dimensional representation leads to information loss, while a higher dimension increases model complexity, making proper dimensionality selection crucial. Since the model's performance remains stable for $h_d = 128, 256, 512$, we choose 128 to reduce model parameters

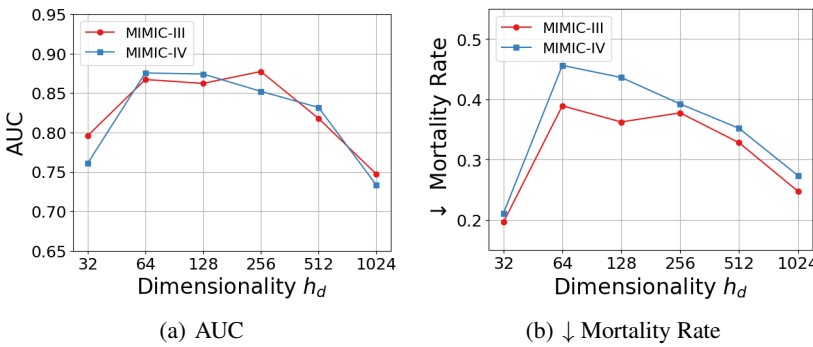

(a) AUC

(b) ↓ Mortality Rate

*Figure 9.* The performance of SAFER under different hidden dimensionality $h_d$

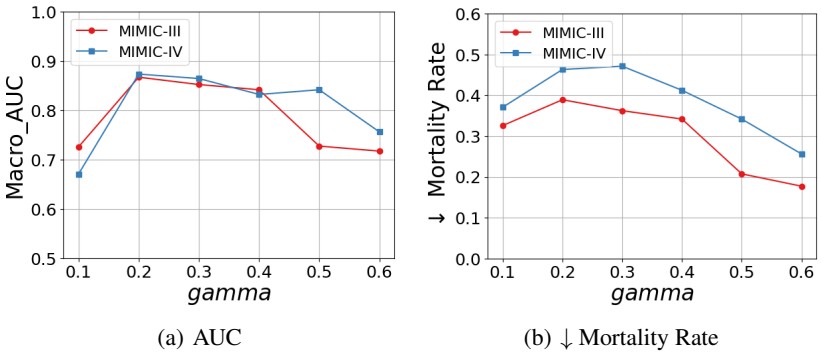

(a) AUC

(b) ↓ Mortality Rate

*Figure 10.* The performance of SAFER under different $\gamma$ value in loss function

and improve computational efficiency.

$\gamma$ **in the Loss Function**: Figure 10 illustrates the performance of SAFER under different $\gamma$ values, guiding the selection of an optimal $\gamma$. A small $\gamma$ leads to decreased performance, confirming the necessity of incorporating this penalty term. However, an excessively large $\gamma$ is also detrimental, as it shifts the model's focus away from the supervised signal during training, ultimately reducing overall performance.

These findings support the stability of SAFER under moderate hyperparameter variation, and highlight the importance of risk-aware fine-tuning in achieving consistent improvements in both predictive accuracy and patient safety.

## C.4. Case Study

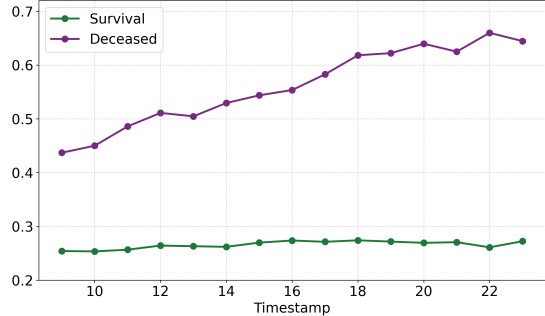

*Figure 11.* Trends in uncertainty scores over time.

To further illustrate the interpretability of SAFER's uncertainty estimates, we select a subset of 10 surviving and 10 deceased

patients with comparable sequence lengths. At each timestamp, we predict the subsequent treatment target using varying historical windows and compute the corresponding uncertainty scores. As shown in Figure 11, the average uncertainty scores for surviving patients remain low and stable throughout the clinical timeline. In contrast, deceased patients exhibit a distinct upward trend in uncertainty as their condition deteriorates.

This divergence reflects a growing difference in predictive stability between the two cohorts. For surviving patients, the model consistently maintains high confidence, likely due to regular disease progression and coherent treatment-response patterns. Conversely, the increasing uncertainty observed among deceased patients suggests a transition into more complex or irregular clinical dynamics, where prediction becomes inherently more difficult.

The elevated uncertainty in deceased trajectories may arise from two primary sources: (1) ambiguous treatment behaviors driven by rapid physiological decline or emergent interventions; and (2) limited representational coverage of similar deteriorating cases in the training distribution, resulting in increased epistemic uncertainty. These observations align with our core modeling assumption that treatment labels for deceased patients are more likely to be noisy or unreliable due to outcome ambiguity and clinical variability.

Notably, these temporal uncertainty trends provide a form of counterfactual interpretability. By capturing the divergence in predictive confidence over time, SAFER not only differentiates between stable and high-risk trajectories but also offers a potential mechanism for proactive clinical risk detection. In practice, this trajectory-level uncertainty signal could be leveraged to identify patients transitioning into unfamiliar or high-risk states, thereby enabling timely human intervention.

In summary, this analysis underscores the utility of SAFER's uncertainty estimates as both a diagnostic and interpretive tool, particularly in high-stakes clinical environments where model trustworthiness and actionable risk awareness are essential.

