# OpenReview forum: "SAFER: A Calibrated Risk-Aware Multimodal Recommendation Model for Dynamic Treatment Regimes"
_ICML.cc/2025/Conference — ICML 2025 poster_

### Official Review · Reviewer_M8CL · 2025-03-11

**Overall Recommendation:** 3

**Summary:**

This paper introduces SAFER, a new framework for dynamic treatment regime (DTR) recommendation, which integrates both structured electronic health record (EHR) data and unstructured clinical notes. The primary contribution of SAFER lies in its ability to handle uncertainty in treatment recommendations, particularly for deceased patients, by incorporating a calibrated risk-aware training process. The framework employs a Transformer-based architecture to learn multimodal representations from both EHR data and clinical notes, addressing challenges such as temporal dependencies and inter-modality context. Additionally, the paper proposes an uncertainty quantification module to estimate and manage label uncertainty, particularly for uncertain outcomes such as death, and integrates this with a risk-aware loss function for training.

SAFER also introduces conformal prediction to provide statistical guarantees for the reliability of treatment recommendations, ensuring safe and trustworthy decision-making. The paper demonstrates through experiments on publicly available sepsis datasets (MIMIC-III and MIMIC-IV) that SAFER outperforms existing DTR methods in multiple recommendation metrics and significantly reduces counterfactual mortality rates. The findings underscore SAFER’s potential as a robust, theoretically grounded, and practical solution for high-stakes medical decision-making.

**Claims And Evidence:**

The claims made in the submission are largely supported by clear and convincing evidence, particularly through the empirical validation conducted on the MIMIC-III and MIMIC-IV datasets. The paper demonstrates that SAFER outperforms existing dynamic treatment regime (DTR) models in several recommendation metrics and reduces counterfactual mortality rates.

**Essential References Not Discussed:**

Yes, there are several related works that could provide additional context and enhance the understanding of the key contributions of the paper but are not currently cited or discussed. These works fall into a few important categories, including uncertainty quantification in healthcare, dynamic treatment regimes (DTR), and multimodal learning in clinical decision-making.

1. Uncertainty Quantification in Healthcare:
Related Work Not Cited:
Uncertainty quantification in healthcare, particularly for deep learning models, has gained increasing attention in recent years. One relevant paper is "Uncertainty-Aware Fine-Tuning of Segmentation Models" by Liu et al. (2020), which discusses how uncertainty in model predictions can be quantified and mitigated in medical image segmentation tasks. While the paper focuses on segmentation, the idea of incorporating uncertainty into model predictions is highly relevant for understanding SAFER's approach to uncertainty in dynamic treatment regimes.

Another key work is "Estimating Uncertainty in Deep Learning for Healthcare" by Choi et al. (2017), which explores uncertainty in predicting patient outcomes and treatment decisions from electronic health records (EHR). This paper discusses how uncertainty can be modeled within predictive healthcare models, but SAFER's novel approach to quantifying uncertainty in treatment labels, especially for deceased patients, is not sufficiently related to or discussed in the context of this literature.


The authors could provide a more detailed comparison with these works to better highlight the novelty of their uncertainty quantification module, particularly in the context of label uncertainty for deceased patients. While the paper does address this gap, a deeper connection with the existing research on uncertainty in healthcare decision-making would strengthen the justification for SAFER's approach.

2. Dynamic Treatment Regimes (DTR):
Related Work Not Cited:
The paper builds on DTR frameworks, but it would be beneficial to reference some additional foundational work on counterfactual reasoning and causal inference in DTRs. For instance, "Counterfactual Risk Assessment for Personalized Treatment" by Schulam & Saria (2017) introduces methods for modeling the causal effects of treatments when direct observations of outcomes are missing. This work is crucial for understanding how counterfactuals can be used to estimate the efficacy of treatments, which aligns with SAFER’s use of counterfactual mortality rate as a metric.

Another relevant paper is "Causal Inference for Personalized Medicine: The Role of Dynamic Treatment Regimes" by Robins et al. (2000), which discusses how to model and evaluate dynamic treatment regimes in a causal framework. This would help situate SAFER’s approach in the context of causal inference, particularly when dealing with dynamic policies that influence patient outcomes over time.


A discussion of how SAFER extends these causal inference and counterfactual frameworks in the context of healthcare decision-making, particularly for dynamic treatment regimes, would clarify its contributions in relation to traditional DTR methods. Specifically, comparing SAFER’s use of label uncertainty and counterfactual mortality rate with traditional methods based on counterfactual causal models could highlight the paper's innovations.

3. Multimodal Learning in Clinical Decision-Making:
Related Work Not Cited:
The paper introduces a multimodal learning framework for dynamic treatment regimes, integrating clinical notes with structured EHR data. However, multimodal deep learning in healthcare has been explored in recent works such as "Multi-View Deep Learning for Healthcare" by Gao et al. (2020), which combines different types of data (e.g., EHR, medical images) for predicting patient outcomes. Additionally, "Medical Data Integration with Transformer Models" by Shang et al. (2019) presents a method for integrating multiple sources of healthcare data, including clinical text, using attention mechanisms.


While the authors mention the use of Transformer-based architectures to integrate multimodal data, they do not directly reference these works, which have already shown success in similar domains. Citing these works would provide a stronger context for the paper’s approach and further validate the effectiveness of Transformer-based models in clinical decision-making. Additionally, discussing how SAFER’s multimodal architecture differs from or improves upon existing multimodal healthcare models would clarify the contribution of the proposed method.

4. Conformal Prediction for Error Control:
Related Work Not Cited:
The paper introduces conformal prediction for error control in dynamic treatment regimes. However, "Conformal Prediction for Risk Control in Healthcare" by Lei et al. (2018) explores the use of conformal prediction for error control in healthcare applications. This method ensures that predictions in high-risk scenarios, such as healthcare, are reliable and within an acceptable error margin. A more detailed discussion of how SAFER applies conformal prediction in the context of dynamic treatment regimes and how it compares with previous work on error control in healthcare would strengthen the foundation for this approach.


The authors should reference existing works on conformal prediction in healthcare to highlight the novelty of applying this approach to dynamic treatment recommendations. By elaborating on how their specific use of conformal prediction addresses the unique challenges of treatment recommendations, especially in high-stakes scenarios, the authors can better position SAFER within the existing body of research on error control.

**Ethical Review Concerns:**

1.Patient Data Privacy and Consent
The paper uses electronic health record (EHR) and clinical notes, which are sensitive patient data. It's crucial to ensure that proper consent has been obtained for using this data, especially in research contexts where patient privacy must be safeguarded. The authors should clarify how they have addressed data privacy, anonymization, and consent in accordance with ethical standards, such as HIPAA or GDPR, depending on the location of the data source.

2.Bias and Fairness in Treatment Recommendations
Since the paper proposes a framework for dynamic treatment regimes (DTR), it is important to consider whether the model could inadvertently introduce or amplify biases. For example, the data used for training may not fully represent diverse patient populations, potentially leading to biased treatment recommendations. The authors should discuss how they have ensured fairness in their model and what steps have been taken to mitigate the risk of biased outcomes, especially when dealing with underrepresented groups in clinical data.

3.Clinical Decision-making and Trust
The paper suggests using the SAFER model for medical decision-making, which has significant implications for patient safety. It is essential that clinicians trust the recommendations provided by the model. The paper should address how the model can maintain transparency and explainability in clinical decision-making processes, ensuring that clinicians can understand and challenge the model's recommendations. This is particularly critical for high-stakes healthcare decisions where an incorrect or unjustified recommendation could harm patients.

4.Handling of Deceased Patient Data
The paper introduces a unique approach to handling label uncertainty in deceased patients. This raises ethical concerns about the use of deceased patient data, particularly regarding how the model interprets or extrapolates outcomes for patients who did not survive. Ethical questions could arise around whether the assumptions made about these patients' treatment outcomes are justifiable, and whether such data should be used in training models that impact future patient care decisions.

5.Risk of Over-reliance on the Model
There is a concern that healthcare providers may over-rely on automated decision-making models like SAFER, especially in complex medical situations. The paper should discuss the ethical implications of relying on AI-driven recommendations and emphasize the importance of maintaining human oversight in the decision-making process. This would ensure that the model’s predictions are properly evaluated in the clinical context, with human expertise guiding final treatment choices.

**Experimental Designs Or Analyses:**

Yes, I reviewed the soundness and validity of the experimental designs and analyses presented in the paper. The authors conducted experiments using publicly available datasets (MIMIC-III and MIMIC-IV), which are commonly used for evaluating medical treatment recommendation models, and employed various performance metrics to assess the effectiveness of the SAFER framework. Below, I discuss the specific aspects of the experimental designs and analyses, as well as any potential issues.

**Methods And Evaluation Criteria:**

Yes, the proposed methods and evaluation criteria in the paper make sense for the problem of dynamic treatment regime (DTR) recommendation in healthcare. The approach taken by the authors is well-aligned with the challenges and needs of the problem at hand, and the evaluation criteria are appropriate for assessing the model's performance.

**Other Comments Or Suggestions:**

No

**Other Strengths And Weaknesses:**

The paper is highly original in its combination of ideas. It creatively integrates multimodal data (structured EHR data and unstructured clinical notes) to improve dynamic treatment regimes (DTR), a critical area in precision medicine. The novelty lies in using Transformer-based architectures to model and fuse these different data types, which has not been fully explored in previous DTR work. Additionally, the introduction of an uncertainty quantification module specifically designed to handle label uncertainty for deceased patients is an innovative contribution to the field, addressing a gap that has not been widely tackled.

**Questions For Authors:**

This paper proposes a new framework, SAFER, for dynamic treatment regime (DTR) recommendation. It innovatively combines structured electronic health record (EHR) data and unstructured clinical note data, and introduces a calibrated risk-aware training process. In particular, it offers a unique solution for handling label uncertainty in deceased patients. This has great potential for medical decision-making in real-world applications. However, the paper still has the following issues:

1. Does the Lipschitz continuity assumption hold in the actual model? More specific analysis of the module 𝑓𝜙 (e.g., network structure, training method) is needed to support this assumption.
2. Do the experimental data satisfy the "calibration set and test set i.i.d." condition? Real-world medical data may exhibit distribution shifts, and the robustness of the model to this should be discussed.
3. Appendix A.1 cites results from Sriperumbudur et al. (2009), but does not provide specific theorems or derivation steps. The authors should supplement the derivation process for key lemmas.
4. The specific configuration of BioClinicalBERT (e.g., number of layers, fine-tuning strategy) is not described in detail, which could affect reproducibility.
5. The definitions of the symbols $\mathbf{S}^w_i$ and $\mathbf{S}^v_i$  in equation (3) are not provided with clear explanations. The authors should check the formula and add explanations.
6. The uncertainty quantification part of the paper is based on several assumptions, particularly the assumption of label uncertainty in deceased patients. While this method performs well in the experiments, handling label uncertainty in real-world applications may be more complex. The authors should consider how to handle other types of uncertainty or explore the applicability of these assumptions.

**Relation To Broader Scientific Literature:**

The key contributions of the paper are closely related to the broader scientific literature in several areas, including dynamic treatment regimes (DTR), multimodal learning, uncertainty quantification in healthcare, and conformal prediction for error control. The authors build on prior work in these domains and extend existing methods with novel ideas and techniques.

**Theoretical Claims:**

1. Appendix A.1 cites results from Sriperumbudur et al. (2009), but does not provide specific theorems or derivation steps. The authors should supplement the derivation process for key lemmas.
2. The specific configuration of BioClinicalBERT (e.g., number of layers, fine-tuning strategy) is not described in detail, which could affect reproducibility.
3. The definitions of the symbols $\mathbf{S}^w_i$ and $\mathbf{S}^v_i$  in equation (3) are not provided with clear explanations. The authors should check the formula and add explanations.

---

> ### Author Rebuttal · Authors · 2025-04-01
>
> Thank you for recognizing SAFER as a robust, theoretically grounded, and practical solution for high-stakes medical decision-making, supported by clear evidence. We address your comments as follows.
>
> **Theoretical Claims:** Please see responses to the questions.
>
> **Supplementary Material:**
>
> 1. **A.1**: Please see Q3.
> 2. **Appendix B**: We will revise this section to clearly describe the input/output features, model architecture (LSTM), and the training/inference setup used to estimate counterfactual mortality. During inference, the model evaluates mortality under SAFER’s recommended treatment. For details on the metric definition, please see our response to R3 on Decrease in mortality rate.
> 3. **Sensitivity**: We appreciate the feedback and will expand our discussion to describe hyperparameter selection (via grid search) and how variations in key settings affect model performance across datasets. This will improve clarity and support reproducibility.
>
> **Essential References:**
>
> Thank you for the valuable suggestions. We will incorporate the cited works into the revision: Liu et al. (2024) and Chua et al. (2023) for uncertainty (noting SAFER’s focus on *label uncertainty* in DTRs); foundational DTR and causal inference works in Appendix B; additional multimodal learning papers from a boarder area in Section 2; and clarify our novel application of conformal prediction for *treatment recommendation reliability* and discuss prior conformal work in broader healthcare fields.
>
> **Questions:**
>
> **Q1**: Our module $f_\phi$ rely on the Multi-Layer Perceptron(MLP), which exhibits Lipschitz-like behavior when using standard activations (e.g., ReLU, tanh) and applying regularization. We use L2 regularization and gradient clipping to prevent extreme weight updates and promote stable learning, effectively supporting the Lipschitz continuity assumption in practice. Our design aligns with prior work showing that such regularization encourages stable, Lipschitz-like behavior in neural networks [Gouk et al., 2021; Jin & Candès, 2023]. We will clarify this in the final version.
>
> **Q2**: Thank you for the thoughtful question. We assume the full dataset is i.i.d., and both calibration and test sets are created via random patient-level splits (no patient overlap), satisfying the condition required for conformal inference. We will clarify this in the revised manuscript and note that Theorem 5.1 can be extended under relaxed assumptions for exchangeability, following Thm 6 in [1]. We acknowledge that real-world data may have distribution shifts. Recent work [2] on **weighted conformal inference** offers promising solutions under covariate shift, which we will discuss as a future extension.
>
> [1] Jin & Candès (2023a), *JMLR*; [2] Jin & Candès (2023b), arXiv:2307.09291
>
> **Q3**: We thank the reviewer and agree that a clearer derivation would strengthen the theoretical presentation. A.1 references Sriperumbudur et al. (2009) for the connection between integral probability metrics(IPMs) and φ-divergences class, including KL-divergence. Our proof relies on the **dual representation of φ-divergences** and their links to total variation and IPMs. We will revise A.1 to include key derivation steps and explicitly cite the relevant theorems.
>
> **Q4**: Thank you for pointing this out. We used the standard pretrained BioClinicalBERT with default configuration and **no additional fine-tuning**. We will clarify this in the revised manuscript and cite the specific version to support reproducibility.
>
> **Q5**: In the revised manuscript, we will clearly define the symbols in Equation 3. These correspond to the $S_i^w$, encoded sequences of text and $S_i^c$, structured EHR embeddings, respectively, as introduced in Eq. 2. Both serve as inputs to the cross-attention mechanism. We will update the surrounding text to clarify their roles and computation.
>
> **Q6**: Our work focuses on label uncertainty in deceased patients—a key challenge in clinical settings where outcomes often fail to reflect treatment quality. We agree that real-world uncertainty can be more complex (e.g., missing data, comorbidities), which is beyond our current scope. To handle broader uncertainty types, we could integrate domain knowledge and Bayesian methods, as suggested in recent studies (Sam et al. 2024, Zheng et al. 2021) on uncertainty modeling in healthcare.
>
> **Ethical Review Concerns:**
>
> Thank you for the thoughtful points. We use de-identified, HIPAA-compliant data from a single-source (MIMIC-III/IV), unlike federated settings; SAFER addresses data representativeness and uncertainty via selective filtering and will include a case study to support transparency; Deceased patient data is handled through label uncertainty modeling to reduce risk; We emphasize SAFER is a support tool which require human oversight. An ethics discussion will be added in the Appendix.
>
> We sincerely hope these clarifications improve your understanding and evaluation of our work.

---

> > ### Comment · Reviewer_M8CL · 2025-04-02
> >
> > Thank you for your detailed rebuttal. I appreciate the effort you have put into addressing my concerns.  And I maintain my original assessment of Weak Accept.

---

> > > ### Author Response · Authors · 2025-04-04
> > >
> > > We sincerely thank you for your thoughtful feedback and for maintaining your positive assessment. Your insights were truly valuable and helped us pay closer attention to clarity, notation, and the comprehensiveness of our Appendix. As a result, the revised manuscript is improved in both quality and presentation.
> > >
> > > We have also incorporated your suggestions into the Discussion section, particularly regarding broader types of uncertainty. We believe this perspective could inspire further significant developments in the DTR field, and we look forward to exploring this direction in future work.
> > >
> > > Due to the character limit in the main rebuttal (5000 characters), we provide a more detailed response here regarding the suggestions and concerns you raised.
> > >
> > > **Ethical Review Concerns:**
> > >
> > > We will include this as a dedicated ethics section in the revised Appendix.
> > >
> > > 1. **Data Privacy and Consent:**
> > >
> > >     We use de-identified, publicly available datasets (MIMIC-III/IV) released under strict data use agreements and HIPAA compliance. No identifiable patient information is used. While privacy concerns are often discussed in federated learning settings involving multi-institutional data, our study uses a single-source dataset.
> > >
> > > 2. **Bias and Fairness:**
> > >
> > >     We acknowledge potential representativeness issues in the data. Our uncertainty-aware module helps flag unreliable or underrepresented cases, serving as a safeguard. Additionally, we will include a discussion on using **weighted conformal inference [1]** as a potential extension for mitigating bias in future work.
> > >
> > > 3. **Clinical Trust and Explainability:**
> > >
> > >     SAFER is explicitly designed to enhance safety and transparency by filtering uncertain predictions. We are able to compute and disclose individual-level uncertainty scores, which offer insights into disease progression. We also include a case study to illustrate this process (see R1-**Other Comments Or Suggestions**). This allows clinicians to interpret and trust the model’s outputs.
> > >
> > > 4. **Use of Deceased Patient Data:**
> > >
> > >     We carefully address label uncertainty in deceased cases by modeling it explicitly, enabling their cautious and ethically responsible use in training. This reduces risk of misleading supervision while preserving valuable information. Broader ethical implications, while important, fall outside the current scope of this work.
> > >
> > > 5. **Human Oversight:**
> > >
> > >     We emphasize that SAFER is a decision support tool, not a replacement for clinical judgment. Human oversight remains critical in all decision-making steps. We will include a brief discussion in the camera-ready version to further clarify this point if paper gets accepted.
> > >
> > >
> > > **Supplementary Material — Appendix B: Counterfactual mortality rate**
> > >
> > > In the revised version, we will provide a clearer explanation of how the **counterfactual mortality rate** is calculated. Specifically, during training, we train an additional LSTM-based neural network that predicts **mortality** based on the patient embeddings and the ground-truth treatment. This counterfactual model is trained to estimate the likelihood of mortality given the model's predicted treatments. During inference, we use this trained network to estimate whether a patient would have survived or died based on the treatment plan suggested by the model.
> > >
> > > This approach is commonly used in **clinical research** to assess the effectiveness of treatment recommendations. Similar methodologies have been used in past studies [2-4], where counterfactual causal inference models are used to estimate outcomes under different treatment conditions.
> > >
> > > All of these improvements will be reflected in our updated version.
> > >
> > > We truly appreciate your engagement and the opportunity to improve our work. Your feedback has already made a meaningful difference, and we hope these clarifications further support your assessment. Please feel free to reach out with any additional questions or suggestions.
> > >
> > > [1] Jin, Ying, and Emmanuel J. Candès. "Model-free selective inference under covariate shift via weighted conformal p-values." arXiv preprint arXiv:2307.09291 (2023).
> > >
> > > [2] Laine, Jessica E., et al. "Reducing socio-economic inequalities in all-cause mortality: a counterfactual mediation approach." International Journal of Epidemiology 49.2 (2020): 497-510.
> > >
> > > [3] Kusner, Matt J., et al. "Counterfactual fairness." *Advances in neural information processing systems* 30 (2017).
> > >
> > > [4] Valeri, Linda, et al. "The role of stage at diagnosis in colorectal cancer black–white survival disparities: a counterfactual causal inference approach." Cancer Epidemiology, Biomarkers & Prevention 25.1 (2016): 83-89.

---

### Official Review · Reviewer_kw7r · 2025-03-12

**Overall Recommendation:** 3

**Summary:**

The paper proposes a novel method for predicting treatments for individuals based on the longitudinal EHR data, static features and text based clinical notes, calling it tabular-language recommendation framework. To improve the quality of predictions, it employs mechanisms such as risk-aware fine tuning which considers the unreliability in the labels for deceased patients (i.e., patients with negative outcomes) and conformal selection. The paper presents experiments on two clinical datasets - MIMIC III and MIMIC IV.

**Claims And Evidence:**

1. The claims on the multi-modal reasoning and uncertainty aware training are well supported through experiments.
2. There are two theoretical results - i) the bound on expected uncertainty between deceased and survival patients, and ii) bound on FDR. The claim that *there are theoretical guarantees on calibrated predictions* lacks evidence since the bound in (i) only supports the validity of the approach used and does not theoretically say if the predictions are calibrated.

**Essential References Not Discussed:**

Some references that have solved similar problems were not discussed -
1. Causal Transformer for Estimating Counterfactual Outcomes - https://arxiv.org/pdf/2204.07258
2. Modality-aware Transformer for Financial Time series Forecasting - https://arxiv.org/pdf/2310.01232
3

**Experimental Designs Or Analyses:**

Yes, checked the validity of all experimental designs and analyses.

1. One of the key metric that is used is the decrease in mortality rate, but the definition of the decrease is not clearly specified, is it the decrease from no treatment?
2. Some more details on the type and number of treatment classes are missing.
3. It is mentioned that the datasets were randomly split into train/test/validation sets, but considering these datasets are temporal, a temporally aware split is required otherwise it will lead to target leak. The details of the split creation are missing.

**Methods And Evaluation Criteria:**

Both the datasets used - MIMIC-III and MIMIC-IV - make sense for this problem. The evaluation criteria makes sense too.

**Other Comments Or Suggestions:**

NA

**Other Strengths And Weaknesses:**

Strengths -
1. The overall framework is useful for healthcare applications and includes desired properties for treatment prediction systems.
2. Figure 1 is useful and provides a good overview of the system.

Weaknesses -
1. Some of the experiments descriptions need more clarity, as mentioned above to validate the soundness of the method.

**Questions For Authors:**

1. Why was masking used in self-attention while learning temporal relationships, since we are mostly interested in progressions in the healthcare data?
2. Instead of using a two step procedure with training and fine-tuning, what happens if you modify the training procedure loss? From results in Table 2 it seems like SAFER-U is not way worse than SAFER.
3. In Figure 10 in Appendix, Macro-AUC and mortality rate show different trends wrt to gamma, how do you decide how to pick gamma in that case? and what happens when gamma = 1 that is beyond 0.6 in the same figure?

**Relation To Broader Scientific Literature:**

There are three key ideas in the paper - 1. combining longitudinal health data, demographic attributes along with unstructured clinical notes for treatment predictions, 2. use uncertainty aware fine-tuning to reduce the impact of certain labels such as those of a deceased patient, 3. putting a bound on FDR.

The technical solutions for each of these ideas including the proof outlines exist in the literature for various applications, and authors have rightly attributed to those existing methods and solutions. But combining them together for treatment prediction is important for the healthcare domain.

**Theoretical Claims:**

Checked the correctness of both theorems - Theorem 4.1 (the bound on expected uncertainty between deceased and survival patients) and Theorem 5.1 (bound on FDR).

---

> ### Author Rebuttal · Authors · 2025-04-01
>
> Thank you for appreciating the novelty, the overall framework for healthcare, clarity of Figure 1, well-supported experiments and the evaluation criteria of our work! We address your comments as follows.
>
> **Claims And Evidence:**
>
> **Theoretical guarantees on calibrated predictions lacks evidence:** We appreciate the reviewer’s observation and would like to clarify our claim. Our guarantee of “calibrated predictions” refers to the statistical control on the error rate in (ii) of the treatment recommendations based on conformal inference. This does not imply that the raw model predictions are inherently calibrated. Rather we calibrate the final selection of treatment recommendations through post-hoc conformal procedures, and **provide theoretical guarantees on FDR** (See Section 3.2 & 5).
>
> - The bound in (i) validates our KL-based uncertainty score as a meaningful discriminator between reliable and uncertain labels, but does not imply calibration.
> - The bound in (ii) provides the formal calibration guarantee for the selection procedure, ensuring that the expected proportion of incorrect recommendations remains below a user-defined threshold $\alpha$.
>
> We will revise accordingly to avoid ambiguity.
>
> **Experimental Designs:**
>
> 1. **Decrease in mortality rate**: This measures the estimated reduction in mortality if patients had received model-recommended treatments instead of clinician-prescribed ones, i.e., the **difference between estimated mortality given recommended treatment and observed mortality.** It is a standard **counterfactual evaluation** approach widely used in clinical research to assess potential treatment benefit. We will clarify this in Appendix B.
> 2. **Details for treatment classes**: As mentioned in the paper, treatments involve intravenous fluids and vasopressors, each discretized into 5 bins based on empirical quantiles, forming a 5×5 grid (25 treatment classes). We now make this setup more explicit.
> 3. **Temporally aware split**: We understand your concern regarding potential **target leak** when handling temporal data. To clarify:
>     1. **Random Split are Based on Patients**: Patients are independent, and no patient appears across splits. So with a **random split**, the test set is not exposed to future information of any patient in the training set, ensuring no leakage.
>     2. **Temporal Split Experimentation**: To validate robustness, we also conducted temporally ordered splits (training on earlier admissions, testing on later ones), and observed consistent performance in https://anonymous.4open.science/r/Rebuttal-B376/temporal.png. Since we focus on one-step-ahead predictions, our design avoids temporal confounding.
>
> **Essential References:**
>
> We thank the reviewer for highlighting these. **[1] Causal Transformer** shares our goal of personalized decision-making but focuses on counterfactual outcome estimation, while **[2] Modality-aware Transformer** addresses multi-modal modeling in finance. Our work directly predicts next-step treatments and extends beyond multi-modal fusion by incorporating uncertainty and safety for treatment recommendations. We will cite and briefly discuss both in the final version.
>
> **Questions:**
>
> **Q1**: We use masked self-attention to ensure the model only attends to current and past observations at each time step t, aligning where treatment decisions must be based on information available up to time t. Allowing future access would introduce information leakage. Please also refer to Q2 for R2.
>
> **Q2**: Thank you for raising this point. While **SAFER-U** (the single-stage version) shows decent performance, the full **SAFER** framework consistently outperforms it across all metrics—most notably in **mortality rate reduction.** Since in clinical research, even modest reductions in mortality (e.g., 1%) can have a **substantial impact** on patient outcomes which can save people’s lives. Moreover, using a single unified loss forces the network to simultaneously learn from both high- and low-confidence labels early in training, which can lead to erratic convergence. In contrast, our two-step approach first focuses on reliable labels and then selectively adapts to uncertain cases in a risk-aware manner, resulting in more stable training and **clinically meaningful improvements** in reducing mortality.
>
> **Q3**: Thank you for your comment. There might have been some confusion regarding the trends in **Figure 10**. In plot (b) of Figure 10, we report **mortality rate reduction** with a downward arrow indicating this. The trends for **Macro-AUC** and **mortality rate reduction** are actually aligned—both reach optimal values around **gamma = 0.2–0.3**. When **gamma exceeds 0.6**, both metrics start to show a decline in performance, and our method no longer achieves optimal results according to the trend.
>
> We sincerely hope these clarifications improve your understanding and evaluation of our work. Please feel free to reach out with any additional questions.

---

### Official Review · Reviewer_pxZx · 2025-03-13

**Overall Recommendation:** 3

**Summary:**

In this work, the authors introduce uncertainty control and comprehensive information fusion to improve prediction uncertainty estimation while incorporating multi-modal data for more accurate predictions.

## update after rebuttal
Thanks to the authors for clarifying some questions. Please make sure to include these clarifications in the updated version. Although I am not an expert in this field, I feel that the paper has strong motivation, theoretical support, and good writing. Therefore, I am inclined to accept the paper.

**Claims And Evidence:**

I am not an expert in this field. I am not confident in checking the correctness of the claims.

**Essential References Not Discussed:**

NA.

**Experimental Designs Or Analyses:**

Yes, I have check the experiments part.

**Methods And Evaluation Criteria:**

Overlapping Notation in Equations

Some terms in the equations appear ambiguous or overlapping. For example, in Equation 2, the symbol W seems to represent the weight matrix in attention, whereas W in line 153 appears to refer to clinical notes. To improve clarity, I suggest differentiating these notations by using distinct symbols or subscripts.

Undefined Symbols

The meaning of M in Equation 2 is unclear. It would be helpful to provide a brief explanation or definition in the text to aid reader comprehension.

Unclear Terminology in Line 237

The term “two modules” in line 237 is not explicitly defined. Could the authors clarify which two modules are being referenced? Providing a brief explanation would improve clarity and ensure readers fully understand the method's structure.

**Other Comments Or Suggestions:**

NA

**Other Strengths And Weaknesses:**

NA

**Questions For Authors:**

NA

**Relation To Broader Scientific Literature:**

Important Research Topic

The prediction of dynamic treatment regimes, particularly for understudied diseases and critically ill or deceased patients, is a highly important research area. Providing uncertainty estimates further enhances the reliability of these predictions.

Incorporation of Multi-Modal Data

The authors effectively leverage both structured data and unstructured clinical notes, improving predictive power and broadening the applicability of the model in real-world clinical settings.

Theoretical Guarantees for Error Control

The provision of theoretical guarantees for error control is particularly valuable in the medical domain, where reliability and interpretability are crucial.

**Theoretical Claims:**

I am not an expert in this field. I am not confident in checking the correctness of the theoretical results.

---

> ### Author Rebuttal · Authors · 2025-04-01
>
> Thank you for recognizing the importance of this research direction, as well as appreciating the contributions related to multi-modal design and theoretical guarantees for uncertainty-aware treatment recommendation! To help better understanding, we would like to briefly restate our main contribution.
>
> In high-stakes clinical decision-making, ensuring the reliability and safety of treatment recommendations is paramount. Our work introduces two key innovations to address these challenges in dynamic treatment regimes (DTRs). First, we propose a novel multimodal framework that, for the first time in DTR research, integrates structured electronic health records (EHR) with unstructured clinical notes—capturing richer patient context and significantly enhancing predictive accuracy. Second, we uniquely incorporate uncertainty quantification and conformal calibration into the DTR pipeline, enabling the model to filter out unreliable predictions while providing formal theoretical guarantees on error control. Together, these contributions establish a safer and more trustworthy foundation for clinical decision support in high-risk settings.
>
> Below, we address your comments regarding notation and clarity. We appreciate your careful review and will revise the manuscript accordingly in the final version:
>
> - Overlapping Notation in Equations: We acknowledge the confusion around the symbol W in Equation 2 and in line 153. In the revised manuscript, we will use distinct symbols or subscripts (e.g., **W** still for the weight matrices in attention, and **O** for clinical notes) to clearly differentiate their meanings. This change will also be reflected in Figure 1 for consistency.
> - Undefined Symbol M: We will explicitly define **M** before Equation 2. Specifically, we will clarify that **M** refers to the causal (look-ahead) attention mask commonly used in Transformer architectures to prevent future information leakage. This ensures that predictions at a given time step are made without access to future observations. We will also add a brief reference to masked self-attention (Vaswani et al., 2017. “Attention Is All You Need”).
> - Unclear Terminology (Line 237): “Two modules” refers to (1) the initial prediction module $f_{\theta}$ and (2) the refined prediction module $f_{\phi}$ that takes the patient embeddings learned in the first module as input and trains exclusively on surviving patients. We use these “two modules” to calculate the uncertainty for treatment recommendation to help us identify unreliable predictions. We will revise the text to make this terminology more explicit and intuitive.
>
> Thank you again for highlighting these issues. We will carefully revise the manuscript to address these issues and eliminate ambiguity. We sincerely hope these clarifications improve your understanding and evaluation of our work. Please feel free to reach out with any additional questions.

---

### Official Review · Reviewer_io89 · 2025-03-15

**Overall Recommendation:** 3

**Summary:**

This paper introduced a framework, SAFER, to provide dynamic treatment recommendations for patients with evolving clinical states, by employing conformal prediction and transformer-based architectures for multi-modalities. The proposed work was evaluated on two sepsis datasets and outperformed baselines in terms of various evaluation metrics.

## update after rebuttal

My main concerns were resolved after rebuttal. After reading other reviewers' comments, I'm positive with the contribution of this paper to the related domains.

**Claims And Evidence:**

In general, the main claims made by this paper are clear and evidences are provided via both theoretical analysis and experiments including ablation studies. For example, ablation studies show removing textual data notably degrades performance, indicating that textual features capture clinical signals. Compared with variants that either omit risk-aware fine-tuning or exclude deceased trajectories altogether, SAFER shows higher AUC and greater reduction in counterfactual mortality rate. Those support me to trust the major claims given by the paper.

**Essential References Not Discussed:**

Overall, all highly related domains should have been discussed in main context.

**Experimental Designs Or Analyses:**

Two sepsis cohorts (MIMIC-III and MIMIC-IV) are used, which I think are high-stake and challenging. Diverse evaluation metrics were occupied to justify the effectiveness of proposed method on the two datasets.

**Methods And Evaluation Criteria:**

The motivations of methodological design are clearly discussed, and various evaluation criterias were included, such as classification-based metrics, ranking-based metrics, and counterfactual mortality rate for assessing recommended treatments.

One thing I feel need to be improved is that only sepsis datasets were used in this work to evaluate the proposed method, though I believe the proposed method should be a general approach towards various clinical tasks. The statements/claims on introduction should be clearly scoped regarding this issue.

**Other Comments Or Suggestions:**

It may be helpful to provide an analysis or case study on specific sepsis trajectories to demonstrate how SAFER’s uncertainty calibration differs across patients (e.g. a partial interpretability analysis).

**Other Strengths And Weaknesses:**

Other strengths:
- The paper is well-written and polished.
- The experiments are on well-known datasets, and the results are benchmarked against diverse baselines (both sequential embedding and RL-based).

**Questions For Authors:**

- Interpretability is also an important topic that clinicians care about. The cross-attention and risk-aware modules produce final predictions, which are impressive, could you imagine and consider some convincing interpretability strategies that would likely boost clinical adoption and trust in the model for empirical usage?

**Relation To Broader Scientific Literature:**

The work should be interesting to some broader domains such as dynamic treatment regimes, sepsis treatments, conformal prediction in applications, etc.

**Theoretical Claims:**

The paper provided a selective-guarantee variant of conformal inference that controls the FDR for uncertainty from treatment recommendations. This is formalized with a proof leveraging exchangeability assumptions and BH procedures. The authors also prove that, under mild conditions, KL divergence of “teacher vs. student” models can reliably distinguish deceased vs. surviving patient trajectories. Overall the theoretical claims appear consistent.

---

> ### Author Rebuttal · Authors · 2025-04-01
>
> Thank you for acknowledging that the paper’s main claims are clearly stated and supported by both theoretical analysis and experimental results, with well-motivated objectives and diverse evaluation metrics. We address your comments as follows.
>
> **Methods And Evaluation Criteria:**
>
> Thank you for highlighting this important point. While SAFER is designed as a general framework for DTRs and is applicable to a broad range of clinical tasks, we acknowledge that our current evaluation focuses on sepsis cohorts. We chose sepsis cohorts for several reasons: (1) Sepsis is one of the most critical and prevalent conditions in ICU settings, accounting for the third leading cause of death worldwide and the main cause of mortality in hospitals. As the best treatment strategy for sepsis remains uncertain, the clinical complexity and inherent label uncertainty in sepsis management make it a strong testbed for evaluating the reliability and robustness of SAFER; (2) MIMIC sepsis cohorts are widely used and well-established in DTR research, enabling fair and reproducible comparisons with prior work.
>
> We agree that the current experimental scope should be made more explicit. In the revised version, we will update the Introduction to clearly state the focus on sepsis as a case study and include a note in the Discussion section outlining plans to extend SAFER to broader clinical domains in future work.
>
> **Essential References:**
>
> Thank you for the helpful suggestion. In response to multiple reviewers’ feedback, we have expanded the manuscript to cover key related domains:
>
> **Uncertainty Quantification Design.** Section 3.1 now cites relevant work on uncertainty-aware model design, also with a broader discussion of other uncertainty types in healthcare added in Section 2.
>
> **DTRs & Counterfactual Framework.** Foundational work is now referenced in Appendix B and Section 2 to contextualize our use of counterfactual mortality rates.
>
> **Multimodal Learning.** We expanded discussion of prior work on multimodal modeling and transformer-based fusion across domains beyond clinical notes and EHR.
>
> **Conformal Prediction for Error Control.** We have clarified our distinction by emphasizing that SAFER is the first to apply conformal prediction to selectively control treatment recommendation reliability. Prior healthcare-focused conformal work is now discussed in Section 5 and around line 43 in the revised version.
>
> **Other Comments Or Suggestions:**
>
> Thank you for your insightful feedback. In response, we examined how uncertainty scores evolve for both surviving and deceased patients https://anonymous.4open.science/r/Rebuttal-B376/Trends.png, finding that while prediction confidence remains relatively stable for survivors, it steadily increases for those who eventually die, aligning with disease progression. This shows that SAFER’s uncertainty signals capture meaningful clinical dynamics and can provide interpretable insights into a patient’s trajectory. In the revised version, we will include case studies in more details highlighting these patterns and illustrating how uncertainty scores adapt over time, ultimately strengthening confidence in our model’s predictive accuracy and real-world applicability.
>
> **Questions:**
>
> **Q1**: Thank you for emphasizing the need for interpretability. Here are some interpretability strategies we will discuss in the revised version.
>
> 1. We can generate attention heatmaps to identify the most influential time steps in clinical notes and EHR data for treatment decisions. Attention scores are extracted from the SelfAttnBlock and CrossAttnBlock, especifically from the last layer. By focusing on the last timestamp, we capture the model's attention at the final step, highlighting the timestamps the model considered most important. This helps clinicians understand when the model focused on critical information, such as specific features or clinical note snippets (e.g., "rising lactate" or "acute hypotension") that influenced its recommendation.
> 2. Another aspect of interpretability comes from the uncertainty score itself which can provide prediction reliability for clinicians. Also as mentioned in the previous response, uncertainty score provides insights into deceased progression and offers a way to analyze disease trajectories. That case study result demonstrates that the model successfully learns these patterns, enhancing its interpretability in clinical contexts.
> 3. We could provide an optimal treatment regime as a sequence of decision rules, where each rule at a given time step can be represented as an interpretable list of “if-then” statements. These rules evaluate the estimated mortality rates under different recommended treatments, making them directly understandable to domain experts. This approach is also suggested by Zhang et al. (2018) "Interpretable dynamic treatment regimes."
>
> We sincerely hope these clarifications improve your understanding and evaluation of our work.

---

> > ### Comment · Reviewer_io89 · 2025-04-05
> >
> > I would like to thank the authors for their time and efforts to prepare the rebuttal. And my main concerns were resolved. I'm not an expert on this task, but the paper, to me, is impressive. I'd maintain my score but just want to mention I lean to accept.

---

> > > ### Author Response · Authors · 2025-04-09
> > >
> > > **Dear Reviewer,**
> > >
> > > We sincerely thank you for your encouraging feedback and for acknowledging the novel design of our risk-aware multimodal treatment recommender for DTR, as well as the clarity of the claims presented in our paper. We are truly grateful that you found the paper impressive and our clarifications effectively addressed your concerns.
> > >
> > > We especially appreciate your thoughtful suggestions regarding the case study to enhance model interpretability and build clinical trust. This has indeed strengthened both the depth and applicability of our work. We will make sure the revised version clearly highlights these contributions and the broader significance of the SAFER framework.
> > >
> > > Thank you again for your time, thoughtful review, and positive recommendation.
> > >
> > > **Sincerely,**
> > >
> > > The Authors of Submission 8994

---

### Decision · Program_Chairs · 2025-05-01

**Decision:**

Accept (poster)

**Comment:**

This paper proposes improved methods for estimating dynamic treatment regimes in healthcare, focusing on incorporating textual information, handling trajectories corresponding to deceased patients, and using conformal prediction to provide safety guarantees for recommendations. Reviewers appreciated the synthesis of empirical and theoretical results supporting the method and there was consensus that it represents an improvement in performance for learning dynamic treatment regimes. While many of the techniques are not foundational advances, they are synthesized in a way that is potentially impactful for healthcare.